# Progress in Electronic, Energy, Biomedical and Environmental Applications of Boron Nitride and MoS_2_ Nanostructures

**DOI:** 10.3390/mi15030349

**Published:** 2024-02-29

**Authors:** Join Uddin, Raksha Dubey, Vinaayak Sivam Balasubramaniam, Jeff Kabel, Vedika Khare, Zohreh Salimi, Sambhawana Sharma, Dongyan Zhang, Yoke Khin Yap

**Affiliations:** 1Department of Physics, Michigan Technological University, 118 Fisher Hall, 1400 Townsend Drive, Houghton, MI 49931, USA; juddin@mtu.edu (J.U.); fraksha@mtu.edu (R.D.); vsivamba@mtu.edu (V.S.B.); zsalimi@mtu.edu (Z.S.);; 2The Elizabeth and Richard Henes Center for Quantum Phenomena, Michigan Technological University, 118 Fisher Hall, 1400 Townsend Drive, Houghton, MI 49931, USA

**Keywords:** electronics, energy, biomedicine, environment, molybdenum disulfide, boron nitride

## Abstract

In this review, we examine recent progress using boron nitride (BN) and molybdenum disulfide (MoS_2_) nanostructures for electronic, energy, biomedical, and environmental applications. The scope of coverage includes zero-, one-, and two-dimensional nanostructures such as BN nanosheets, BN nanotubes, BN quantum dots, MoS_2_ nanosheets, and MoS_2_ quantum dots. These materials have sizable bandgaps, differentiating them from other metallic nanostructures or small-bandgap materials. We observed two interesting trends: (1) an increase in applications that use heterogeneous materials by combining BN and MoS_2_ nanostructures with other nanomaterials, and (2) strong research interest in environmental applications. Last, we encourage researchers to study how to remove nanomaterials from air, soil, and water contaminated with nanomaterials. As nanotechnology proceeds into various applications, environmental contamination is inevitable and must be addressed. Otherwise, nanomaterials will go into our food chain much like microplastics.

## 1. Introduction

The field of nanotechnology has been evolving over the last four decades. Many pioneer works of quantum dots (QDs) [1,2,3] and carbon nanotubes (CNTs) [4] focus on the synthesis and characterization. The more recent works are moving towards practical applications and environmental studies. In the U.S., the National Nanotechnology Initiative (NNI) launched in 2000 focuses on the synthesis, characterization, education, and application of nanomaterials [5]. The revised NNI published in 2011 has expanded the focus into societal and ethical aspects, such as “Safe and sustainable development” and “Societal benefits” [6]. The latest NNI Strategic Plan 2021 has five goals, further emphasizing the promotion of the ”commercialization of nanotechnology”, and the “responsible development of nanotechnology” [7]. Apparently, nanomaterials should consider not just science and engineering but also work toward applications that benefit society. Moreover, nanotechnology should not introduce potential threats to the environment and health. Our research philosophy aligns with the NNI goals. For instance, we have inspired many researchers to use nanotubes for water purification in 2011 [8]. Given the potential water contamination with nanomaterials, we have further developed a simple, universal, and effective method to remove nanomaterials (nanotubes, graphene, BN sheets, ZnO nanowires) from waters that are unintentionally contaminated during the application of nanomaterials [9,10].

Aligning with the goals of NNI, this review article serves two purposes. First, to disseminate recent progress in using nanomaterials that will benefit society, such as electronic, energy (Section 2), biomedical (Section 3) and environmental (Section 4) applications. We specifically focus on the application of boron nitride (BN) and molybdenum disulfide (MoS_2_) nanostructures with sizable bandgaps to distinguish them from the focus on CNTs, graphene, and MXene, which are metallic (except some single-walled CNTs). Secondly, we intended to promote scientists’ awareness of nanotechnology’s environmental impacts. We hope this will encourage scientists to perform research with high ethical value without negatively impacting the environment and living things. As illustrated in Figure 1, the synthesis and application of nanomaterials will contaminate air, soil, and water. Therefore, studying the environmental application of nanomaterials and the hot topics in electronics, energy, and biomedicine is equally crucial. We also like to point readers to other review articles on the applications of BN [11,12,13,14] and MoS_2_ [15,16,17,18] nanostructures. The structural, physical and chemical properties of BN [19,20] and MoS_2_ [16,17] are described in prior publications. The appearance of these nanostructures is schematically illustrated in Figure 2. MoS_2_ could appear in 1T, 2H, and 3R stackings, and c is the thickness of a unit cell.

## 2. Electronic and Energy Applications

### 2.1. Electronic Applications of BN Nanostructures

Recent advancements in electronic applications of hexagonal boron nitride (h-BN) sheets [14,21,22,23,24] and boron nitride nanotubes (BNNTs) [12,13,25,26] suggest that BN nanostructures are promising materials for a wide range of nanoelectronics applications. Insulating h-BN, which has a dielectric constant of 3–4 and a breakdown field of 8–12 MV cm^−1^ [27], is presently considered an ideal substrate and dielectric for 2D and field-effect transistor (FET) devices due to its wide bandgap, atomically flat surface, chemical stability, and improved carrier mobility [28]. All these properties offer opportunities for extending Moore’s law towards high-density integrated circuits [29]. On the other hand, the one-dimensional (1D) BNNTs are electrically insulating substrates that create quantum materials on their tubular surfaces and inside the tubular channels without interfering with the transport properties of the quantum structures [12,13,26,27].

#### 2.1.1. Nanoelectronics Devices

Researchers emphasize using h-BN sheets as a ubiquitous substrate for 2D semiconductors. Ma et al. reported using h-BN sheets as the dielectric layer in MoS_2_ FETs. For this, an h-BN sheet is deposited onto a SiO_2_/Si substrate to reduce the amount of electron doping in MoS_2_ [29]. Yang et al. reported the formation of FETs based on Te nanobelts using a global bottom-gate structure on the h-BN/SiO_2_/Si substrate, as shown in Figure 3a [30]. The van der Waals (vdW) h-BN sheet works as the dielectric substrate by providing an ultra-flat surface free of dangling bonds and reducing scattering centers at the channel material–dielectric layer interface to form high-performance p-type FETs with a high hole mobility up to 1370 cm^2^ V^−1^ s^−1^ at room temperature (Figure 3b) [30]. The local bottom-gate design (Figure 3c) allows good control of channel material. Still, the mobility is lower at 608 cm^2^ V^−1^ s^−1^ (Figure 3d), owing to scattering centers formed after the wet-transfer fabrication process.

Transition metal dichalcogenides (TMDs) are remarkable 2D semiconductors due to their optoelectronic properties tunable by MX_n_ composition, where M is a transition-metal atom (Mo, W, etc.), and X is a chalcogen atom (S, Se, or Te) [15,16,17]. The high contact resistance and surface contamination limit the practical use of TMDs in devices. Phan et al. overcame this problem by introducing a monolayer of h-BN sheet at the interface between chromium (Cr) and tungsten disulfide (WS_2_) in WS_2_ FETs [31]. They reported that metal–insulator–semiconductor (MIS) contacts have a contact resistance ten-fold lower than MS (metal–semiconductor) contacts. The electron mobility is up to 115 cm^2^ V^−1^ s^−1^ at 300 K, which is ten times greater than devices with MS contacts. The authors attributed the enhanced mobility to using h-BN inserted between Cr and WS_2_, which decreases the Schottky barrier [31]. In another study, Iqbal et al. discussed that FETs based on monolayered tungsten disulfide (WS_2_) are low-mobility, and the environment affects the transport properties. They overcame such issues by using h-BN films as the capping layers for the FETs [32]. They fabricated the device using the h-BN/WS_2_/h-BN structure. They observed a high mobility of 214 cm^2^ V^−1^ s^−1^ at 300 K and 486 cm^2^ V^−1^ s^−1^ at 5 K, as well as an ON/OFF ratio of output current of 10^7^ at room temperature, which is free of charged impurities in comparison to SL-WS_2_ FET on SiO_2_ [32]. Ultraclean h-BN encapsulation conserves interface quality and inherent charge transport behavior in semiconductor devices. BN-encapsulated 2D FETs provide device quality analogous to FETs fabricated by dry-transfer fabrication. Such FETs with WSe_2_ and MoS_2_ channels exhibit a low scanning hysteresis of 2 mV, a low interfacial charged impurity density of 10^11^ cm^−2^, and high charge mobilities exceeding 1000 cm^2^ V^−1^ s^−1^ at low temperatures [33].

Boron nitride also helps stabilize 1D vdW channels for high-performance FETs. In this case, BNNTs encapsulate Te atomic chains inside their tubular channel, as illustrated in Figure 4a. Qin et al. filled BNNTs with as few as five Te atomic chains to form a bundle of Te nanowires as thin as 2 nm to 5 nm (Figure 4b–e), which would not be stable without the BNNT shell [25]. Te atoms within each atomic chain are covalently bonded with vdW forces between atomic chains. These bundles of Te atomic chains are quantum materials [12]. As shown in Figure 4f, the frequencies of the in-plane stretching mode (*E*_2*g*_) and the out-of-plan vibrational mode (*A*_1*g*_) decrease as the thickness of the bundle increases. For 2D materials such as MoS_2_, the *A*_1*g*_ increases with the thickness of the material due to the enhanced restoring force between interlayer forces in MoS_2_ [16]. The increase in the *A_1g_* mode in thinner Te atomic chains suggests that the vdW interaction between thin atomic chains is fragile and will not exist without being encapsulated in BNNTs. The authors also constructed FETs, as illustrated in Figure 4g,h. The shells of the BNNTs are plasma-etched to enable electron tunneling between the Te atomic chains and the Ni/Au electrodes. A current density as high as 1.5 × 10^8^ A cm^−2^ is higher than most semiconducting nanowires despite a diameter of 2 nm [25]. The electrically insulating nature of BNNTs has enabled the formation of FETs out of the encapsulated Te atomic chain. Current will flow through the carbon shells if Te is encapsulated within CNTs.

#### 2.1.2. Photovoltaic/Solar Cells

Energy consumption is on the rise worldwide, yet primary energy sources like fossil fuels have detrimental effects on the environment and people’s health due to their emissions of greenhouse gases and other pollutants into the atmosphere [34]. Since solar energy is the cleanest, safest, and most plentiful, solar cells are considered one of the most possible solutions to the world’s energy problem [35]. Yet, from the perspective of materials engineering, some issues concerning device design still need to be resolved. The application of 2D h-BN sheets have attracted significant attention in solar cells to passivate active material surfaces and heterojunction interfaces, as they are layered structures without dangling bonds [36,37]. Tavakoli et al. [38] studied chemical vapor deposition (CVD) grown monolayer h-BN as an effective electron blocking layer (EBL) for organic photovoltaics (OPVs); solar cells using ITO/ZnO/Blend/h-BN/Ag and ITO/h-BN/Blend/ZnO nanocrystals/Ag structures were used for the inverted and conventional structure (Figure 5a and Figure 5d, respectively). These device structures were confirmed using a cross-sectional Scanning Electron Microscopy (SEM) image (Figure 5b and Figure 5e, respectively). They reported that h-BN could substitute hole transport layers (HTLs) with greater shelf-life, operational and thermal stability than devices with MoO_3_ or poly(3,4-ethylenedioxythiophene) polystyrene sulfonate (PEDOT:PSS). For the inverted device structure, the solar-to-power conversion efficiency of devices with h-BN is 6.13%, which is comparable to devices based on MoO_3_ (7.3%) and PEDOT:PSS (7.6%), as shown in Figure 5c. For the conventional device structure, the solar-to-power conversion efficiency of devices (Figure 5f) with h-BN and current density is 4.8%, approaching that of devices with PEDOT:PSS (6.85%).

Cho et al. investigated the impact of the h-BN layer’s passivation and encapsulation properties on the performance and stability of a MoS_2_/WSe_2_ van der Waals heterojunction solar cell [39]. They found that the power conversion efficiency of a solar cell increased by approximately 74%, resulting from a significant rise in both short circuit current, J_SC,_ and open-circuit voltage, V_OC_, due to an overall decrease in the recombination rate occurring at the interface and surface of non-overlapping regions of the semiconductor areas. Also, they reported that the passivated 2D solar cell degraded twice as slowly as the non-passivated cell due to the h-BN top layers. In another study, Raj et al. studied the passivation effect of h-BN, grown via metalorganic chemical vapor deposition, on ITO/i-InP/p^+^-InP solar cells [40]. They reported that by using a few monolayers of h-BN as the passivation layer, the dark current and the reverse leakage current could be reduced by orders of magnitude, resulting in improved solar cell performance ascribed to direct quantum mechanical tunneling and interfacial charge transfer along with a thicker h-BN film, improving passivation and open-circuit voltage, V_oc_; however, a thinner film is required for high short-circuit current, J_sc_.

### 2.2. Energy Applications of BN Nanostructures

#### 2.2.1. Separators in Batteries

Environmental problems on a global scale have increased the worldwide demands for renewables, including solar, wind power, and green energy storage units, to reduce the pollution due to the use of fossil fuels [36,41]. All renewable energy sources need batteries for energy storage devices, prompting researchers to explore new materials to improve traditional battery design [42]. Among them, h-BN sheets and BNNTs are employed in various energy applications, including but not limited to battery electrodes and electrolytes, due to their superior mechanical properties, thermal conductivity, and chemical stabilities [43,44]. Lithium-ion batteries (LIBs) are prone to safety issues due to initial cell overheating caused by short circuits in high-temperature, high-current atmospheres. Therefore, Rahman et al. investigated the spread-coating of poly (vinyl alcohol)/BNNTs on polyolefin (PP) separators with the doctor blade technique without impeding Li^+^-ion diffusion within porous channels [45]. They found that the BNNT separator is more stable at temperatures up to 150 °C. The BNNT separator shows better electrolyte wettability and enhanced electrochemical performance, which prevents internal short circuits at high temperatures and reduces LIB thermal runway failure. In another study, Moraes et al. studied dry phase-inversion-prepared free-standing LIB composite separators made of carbon-coated hexagonal boron nitride (h-BN) nanosheets and a poly (vinylidene fluoride) (PVDF) polymer that does not require stretching, etching, or immersion in coagulation baths [46]. As a result, carbon-coated h-BN nanosheets in the PVDF matrix enhance essential interfacial interaction during phase inversion, resulting in porous, flexible, free-standing composite separators that provide safe lithium-ion battery operation at a high temperature of 120 °C, compared to conventional polyolefin separators, thus improving the safety and high-temperature performance of rechargeable LIB technology.

#### 2.2.2. Energy Storage in Batteries

LIBs are fast reaching their theoretical limits, with energy densities ranging from 240 to 420 Wh/kg. Consequently, next-generation batteries will need greater energy densities and longer lifespans [47]. Lithium–sulfur batteries (LSBs) are a promising alternative to LIBs because of their high theoretical energy density, low price, ecological compatibility, and abundance of sulfur in the earth [48]. Despite the many potential uses for LSBs, this technology still faces a few challenges. For example, the shuttle effect [49] and cycle stability are reduced by short circuits caused by lithium metal dendrite formation [42]. Kim et al. reported that BNNT separators reduce the shuttle effect at the cathode and impede dendrite growth on the Li metal anode through pore exclusion and polysulfide interaction, specifically by incorporating pure BNNT (p-BNNT) into lithium–sulfur battery cells [42]. A higher energy density of 1429 mAh/g is possible and stable over 200 cycles because the p−BNNT separator increases lithium ionic conductivity, diffusivity, and stability, compared to the polypropylene separator in which LSBs cell stopped working after 155 cycles [42]. Mussa et al. reported that integrating h-BN sheets and graphene could prevent shuttle effects. It was found that adding h-BN to reduced-graphene oxide improved the electrochemical performance of graphene-based Li-S batteries by 2.5 times [50].

Furthermore, aluminum–sulfur batteries (Al-S batteries) are one of the most viable alternatives to lithium-ion batteries because of their low cost, earth abundance, electrochemical efficiency, lack of flammability, and high theoretical volumetric and gravimetric energy densities (3177 Wh L^−1^ and 1392 Wh kg^−1^) [51]. Zhang et al. demonstrated for the first time that S and sulfur compounds may act as fixers during repeated charge/discharge cycles in two-dimensional layered materials such as MoS_2_, WS_2_, and BN and showed that BN bonds with S and/or sulfide compounds to preserve S capabilities from deterioration for aluminum–sulfur batteries [52]. They reported for MoS_2_ an immediate decrease from 553 to 100 mAh g^−1^ after 20 charge/discharge cycles; also after 50 cycles, the capacity is reduced to 50 mAh g^−1^. Likewise, for the WS_2_/S/C sample, after 25 charge/discharge cycles, starting discharge capacity drops from 526 mAh g^−1^ to 54 mAh g^−1^. Despite having a comparable beginning capacity for BN/S/C compared to MoS_2_/S/C and WS_2_/S/C, BN/S/C has an outstanding capacity of 532 mAh g^−1^ and Coulombic efficiency of 94.3% along with a remarkable 300-cycle longevity for Al-S batteries, and a discharge voltage plateau of ~1.15 V vs. AlCl_4_^−^/Al (Figure 6a–e).

#### 2.2.3. Hydrogen Evolution Reaction (HER) and Water Splitting

The free energy of atomic hydrogen bonding to MoS_2_ edges has been previously reported to be close to that of conventional platinum. Pristine MoS_2_ has a low catalytic performance for a low concentration of exposed catalytic sites [53]. It is thus essential to focus on engineering MoS_2_ to increase the number of catalytic sites. One strategy is to reduce the size of the MoS_2_ drastically and use MoS_2_ quantum dots as the electrocatalyst. We recently discussed this topic in our review of MoS_2_ quantum dots [15]. In addition to size reduction, other methods to increase the catalytic effect include phase and structural engineering [54,55], catalytic site engineering [56,57], and doping [58]. For a broader look at the electrocatalytic potential of MoS_2_ for hydrogen production, we recommend reading the roadmap cited here [53]. 

Many metal-based photocatalysts can only be activated by UV radiation, and those materials which can absorb visible light frequently experience stability problems [59]. For this reason, recent years have seen increased interest in separating water into oxygen and hydrogen using metal-free photocatalysts [59,60]. Li et al. reported first-principles calculations of semihydrogenated BN sheets (sh-BN) as a metal-free photocatalyst to extract hydrogen from water as irradiated with visible light without using transition metals [59]. They found that sh-BN has a strip-like antiferromagnetic ground state and becomes a semiconductor with a bandgap of 2.24 eV. The redox potentials of H_2_O are well located within the bandgap. On the other hand, a recent experiment suggests that 2D stacking of graphene and h-BN layers offer interfaces with electrochemical active sites [60]. This stacking arrangement is stable over the long run, which allows hydrogen evolution with an overpotential of 28 mV at 10 mA/cm^2^ and oxygen evolution with an overpotential of 360 mV at 50 mA/cm^2^.

### 2.3. Electronic Applications of MoS_2_ Nanostructures

#### 2.3.1. Gas Sensors

Metal oxides are the best for gas-sensing applications regarding sensitivity and resistivity. However, they require a high operating temperature, which leads to increased power consumption. A novel sensing material should possess better sensitivity, stability, and selectivity at room temperature [61]. Schedin et al. fabricated gas sensors using 2D graphene operating at room temperature and paved the way for exploring other 2D materials to enhance the performance of gas sensors [62].

MoS_2_ shows a higher sensitivity in detecting NO_2_ than NH_3_ gas molecules due to the increase in resistance of n-MoS_2_ after absorption of NO_2_. The oxidizing nature of NO_2_ with the unpaired electron of the N atom shifts the Fermi level of MoS_2_ by removing an electron. For NH_3_, the lone pair adds electrons to MoS_2_, and there is no considerable shift in Fermi level [63]. Apart from this, MoS_2_ edge sites and defect sites assisted in the electron transport of gas molecules due to high electron density in the d-orbital [64]. However, when pristine MoS_2_ is exposed to the air environment, it absorbs more oxygen, leading to the device’s instability. This can be overcome by decorating nanoparticles on the surface of MoS_2_ to improve stability, selectivity and sensitivity. For example, SnO_2_ nanoparticles (NPs) decorated on MoS_2_ nanosheets showed an excellent selectivity towards NO_2_ gas, complete recovery at room temperature and faster response. This is due to the increased work function of SnO_2_ and acts as a passivation layer by preventing oxygen interaction [65]. Au NPs on MoS_2_ demonstrated high performance in detecting NH_3_ at 60 °C. This is due to the high catalytic activity and the high probability of the NH_3_ molecule interacting with MoS_2_ [66].

#### 2.3.2. Photodetectors and Phototransistors

Two-dimensional MoS_2_ is popular as a photodetection material due to its wide absorption range, high current on/off ratio, tunable bandgap, chemical stability, and good carrier mobility. MoS_2_ shows significant characteristics in the visible region of the spectrum, and its performance can be improved by incorporating NPs and QDs, leading to better detection in near UV and IR spectra, and heterostructures with other 2D materials can enhance the response time [67].

Au–MoS_2_–ITO exhibited a fast photo response of ~64 μs in the visible and IR spectra and excellent sensitivities to photocurrent due to the vertical Schottky junction [68]. The MoS_2_−graphene−WSe_2_ heterostructure, where gapless graphene was inserted between the p-n junction, showed wider photodetection from 400 to 2400 nm and faster charge transport. This device performed an efficient charge carrier separation due to the built-in electric field and achieved a faster rise/fall time of 53.6/30.3 μs [69]. Few-layered MoS_2_ on SiO_2_ nanowires exhibited an absorption range from 280 to 1200 nm with an excellent response time 2.4/7.3 µs and a detectivity of 5.3 × 10^12^ Jones due to the high-quality heterojunction [70]. Au–few-layered MoS_2_–Si/SiO_2_ showed a good response time (rise = 70 μs, fall = 110 μs) in the visible region and responsivity R of 0.57 A/W. The fast response time is due to the interdigitated fashion of the Au electrode [71]. Multilayer MoS_2_ is protected from the environment by encapsulation between ITO and copper oxide (Cu_2_O). The authors reported a detectivity of 3.2 × 10^14^ Jones and responsivity of 5.77 × 10^14^ A W^−1^ at an incident power density of 0.26 W m^−2^ at an external bias of −0.5 V. This device could even detect > 10^4^ Jones at zero bias due to the built-in electric field [72].

#### 2.3.3. Electronic Devices

Continuous miniaturization of Si-based FETs of less than 10 nm has reached its limitation in carrier mobility [73]. One could use atomically thin 2D materials with good mobility for future FETs. Among 2D materials, MoS_2_ has exhibited a high I_On_/I_Off_ ratio and low subthreshold swing [74]. However, MoS_2_ has limitations where the contact interface suffers from the Schottky barrier and Fermi-level pinning. These inherent characteristics significantly form a high contact resistance with the electrode material.

Fermi-level pinning is due to the metal-induced gap states (MIGS), which form an energy barrier at the metal-semiconductor interface and lead to high contact resistance. Shen et al. reported using semimetallic bismuth (Bi) as the contact electrode on monolayer MoS_2_. The authors achieve a zero Schottky barrier height through this approach, a low contact resistance of 123 Ω μm, and a high on-state current density of 1135 mA per μm. The semimetallic Bi has near zero density-of-state at the Fermi level, where few MIGS can be induced. As a result, MIGS are purely contributed by the valence band and thus can be filled up and saturated. Therefore, MoS_2_ will be in a degenerate state and free of a Schottky barrier at the contact interface [75]. The authors confirm the gap-state saturation of Bi- MoS_2_ by first-principles calculation based on the crystal structure identified from TEM. The selected-areas electron diffraction suggests that the Bi (0001) plane is parallel to the plane of MoS_2_, as utilized for the calculation as schematically illustrated in Figure 7.

Recently, a single-crystal SrTiO_3_ (STO) is used as the top-gate dielectric for high-performance FETs based on n-type MoS_2_ and p-type WSe_2_ [76]. In the case of MoS_2_, few-layer graphene was used as the source and drain electrodes. Such devices showed one of the best subthreshold swings of 66 mV dec^−1^, approaching the theoretical value of 60 mV dec^−1^. The authors also demonstrated a high on/off ratio of 108 due to the strong electrostatic modulation, where SrTiO_3_ served as the gate dielectric with a high dielectric constant (*κ*). Huang et al. also reported MoS_2_ FETs coupled with sub-10-nm capacitance equivalent thickness (CET) of perovskite SrTiO_3_. The FETs exhibited a subthreshold swing of 71.5 mV dec^−1^ [77]. The authors attributed the performance to the effective vdW gap as the tunneling barrier that reduces leakage current and carrier tunneling from the high-*κ* perovskite.

### 2.4. Energy Applications of MoS_2_ Nanostructures

#### 2.4.1. Lithium-Ion Batteries

LIBs with graphite-based material as the anode have a low theoretical capacity, which limits their application for high-density energy storage. To improve the performance of LIBs, scientists have started looking for novel materials for electrodes that offer high energy density storage, long life, and are environmentally friendly. MoS_2_ has a higher theoretical capacity (670 mAhg^−1^) than graphite (372 mAhg^−1^). The layered MoS_2_ has a larger interlayer distance (~0.62 nm), allowing faster intercalation and diffusion without changing its volume.

Teng et al. grew MoS_2_ nanosheets vertically on graphene sheets and achieved a stable cyclic performance of 1077 mAhg^−1^ at 100 mAg^−1^ after 150 cycles and a long cycle life of 907 mAhg^−1^ at 1000 mAg^−1^ after 400 cycles [78]. The authors attributed such performance to the large number of active edges in the MoS_2_ nanosheets and the interfacial interaction of C-O-Mo bonds, which enhances the electron transportation rate. On the other hand, metallic 1T MoS_2_ nanosheets were vertically grown on the surface of graphene with a large interlayer distance of 0.98 nm. The 1 T MoS_2_ structure provides expanded interlayer space with a more active area for lithium-ion storage and exhibits excellent performance with a high capacity of 666 mAh g^−1^ at a high current density of 3500 mA g^−1^ [79]. Li et al. reported carbon-free anodes based on a composite of MoS_2_ nanosheets–NiMoO_4_ nanowires. The LIBs deliver a mass-specific capacity of 940.1 mAh g^−1^ and a discharge capacity of 941.6 mAh g^−1^ after 750 cycles [80]. We encourage readers to read a recent review article on using composites of MoS_2_ for energy-storage applications [81]. 

#### 2.4.2. Sodium-Ion Batteries

Sodium-ion batteries (SIBs) have attracted a great attention due to their low redox potential, high energy density, long cycling stability and high capacity [81]. Sodium ions have a larger atomic radius (1.02 Å) when compared to lithium ions (0.76 A) [82], so anode material should possess larger interlayer spacing to accommodate Na^+^ ions. Two-dimensional MoS_2_ can store Na^+^ ions due to its large surface area but suffers from poor cycling stability.

A metallic 1 T MoS_2_ sandwich grown on graphene was used as the anode for SIBs [83]. These SIBs exhibit a high reversible capacity of 313 mAh g^−1^ at a current density of 0.05 A g^−1^ after 200 cycles and a high rate capability of 175 mAh g^−1^ at 2 A g^−1^. This anode prevents the aggregation of MoS_2_ and exhibits good electrical conductivity, reduced ion diffusion length, and good cyclic stability [83]. Few-layer MoS_2_ nanosheets were grown on reduced graphene oxide cross-linked hollow carbon spheres (MoS_2_-rGO/HCS) that form a 3D network [84]. The MoS_2_-rGO/HCS hollow spheres were tested as anodes in both LIBs and SIBs. For LIBs, MoS_2_-rGO/HCS can deliver a reversible capacity of 1145 mAh g^−1^ after 100 cycles at 0.1 A g^−1^ and a reversible capacity of 753 mAh g^−1^ over 1000 cycles at 2 A g^−1^. For SIBs, the as-developed MoS_2_-rGO/HCS can also maintain a reversible capacity of 443 mAh g^−1^ at 1 A g^−1^ after 500 cycles. This performance is attributed to the 3D porous structures with expanded interlayers that provide a shorter ion diffusion path and improved ion diffusion mobility [84]. Recently, 2D lamellar stacked nanosheet VS_2_/MoS_2_ heterostructures were tested for SIBs [85]. The authors demonstrated SIBs with an excellent capability of 644 mAh g^−1^ at 10 A g^−1^ and long life cycle stability of 454.5 mAh g^−1^ at 2 Ag^−1^ after 1000 cycles. The electron density is higher in the VS_2_/MoS_2_ composites, facilitating Na^+^ adsorption and enhancing the reaction kinetics. In addition, lattice disorder at the interface contributes towards stability [85]. On the other hand, bundled defect-rich MoS_2_ displayed a high reverse capacity of 350 mAh g^−1^ at 2 A g^−1^ and 272 mAh g^−1^ at 5 A g^−1^ after 1000 cycles [86]. The vacancies assisted in the diffusion of MoS_2_ in three dimensions rather than on basal planes and improved the kinetics. This architecture prevents the stacking of MoS_2_ and, therefore, enhances stability.

#### 2.4.3. Supercapacitors

Supercapacitors are a promising candidate for energy storage devices for their high power density, high energy density, quick charge and discharge rates [87]. Carbon-based materials and transition metals are extensively used as electrodes due to their excellent electrical properties, mechanical strength, and light weight. Among them, layered structures of MoS_2_ have gained considerable attention as the electrodes in supercapacitors because of their high flexibility, high electrical conductivity, and large surface-to-volume ratio. The layered structure of MoS_2_ facilitates double-layer charge storage devices due to its fast ionic conductivity [88]. A composite of MoS_2_ and carbon-based material is an efficient way to enhance conductivity and electrochemical properties [89,90,91,92,93]. 

Three-dimensional nanospheres of MoS_2_ nanosheets were fabricated with a SiO_2_ nanosphere as the template. The nanospheres deliver a specific capacity of 683 F g^−1^ at 1 Ag^−1^ and retain more than 85.1% after 10,000 cycles [94]. The authors obtained an energy density of 20.42 Wh/Kg at a power density of 750.31 KW/Kg. This is attributed to the nanospherical morphology of MoS_2_, which increases the surface area for charge transfer. On the other hand, MoS_2_-graphene hybrid nanostructures were synthesized by a facile one-pot chemical method [95]. Electrodes based on these hybrid nanostructures reveal a specific capacity of 756 F g^−1^ at 0.5 A g^−1^ and retain 88% of the initial capacitance after 10,000 cycles. Such a performance is due to the synergistic effects of graphene and MoS_2_ and the uniform morphology of the honeycomb structure of the nanosheet [95]. Flower-like MoS_2_ microstructures synthesized on 3D graphene (3DG/MoS_2_, Figure 8) [93] also exhibited excellent performance with a high specific capacitance of 410 Fg^−1^ at a current density of 1 Ag^−1^ and cyclic stability with retention of 80.3% of capacitance after 10,000 cycles; this is due to sufficient active sites in the flower-like structure for charge transfer and reduced diffusion length [93].

#### 2.4.4. Solar Cells

Two-dimensional MoS_2_ has interesting optoelectronic properties, a tunable bandgap, and a higher absorption coefficient of 10^5^ cm^−1^ throughout the solar spectrum. Monolayer MoS_2_ can absorb up to 5–10% of incident sunlight in the visible spectrum, which is one order of magnitude higher than the absorbance of GaAs and Si [96]. In dye-sensitized solar cells (DSSCs), Pt is widely used as a counter electrode (CE), as it decomposes to PtI_4_ by an I^−^/I^3−^ redox couple, affecting the cell’s performance. MoS_2_ is an excellent alternative to replace expensive Pt and indium tin oxide (ITO) electrodes in thin film solar cells [97]. 

CoS_2_ coated with MoS_2_ as a CE demonstrated a higher power conversion efficiency (PCE) of 6.6% due to the synergistic effect of Co3S_4_ and the transfer of electrons at the interface [98]. Using a 1 T MoS_2_ flower structure as CE in DSSCs exhibited an excellent power conversion efficiency of 7.08% due to the high conductivity and electrocatalytic activity of the 1T phase [99]. Lei et al. studied three different morphologies of MoS_2_ (i.e., multilayered, few-layered, and nanoparticles) as CEs in DSSCs [100]. The highest efficiency was in the order of nanoparticles, multilayered and few-layered. This revealed that catalytic active sites lie on the edges rather than basal planes of MoS_2_. Yuan et al. synthesized MoS_2_ nanosheets on reduced graphene oxide as a CE and led to a PCE of 6.82%, which is higher than that of conventional devices with Pt (6.44%). The MoS_2_ CE enhanced electrocatalytic activity and reduced redox couples [101].

In organometallic-halide perovskite solar cells, as shown in Figure 9, MoS_2_ nanoflakes were used as a buffer that achieved a PCE of 14.9%. MoS_2_ acts as a protective layer to enhance the device stability and as an additional HTL to enhance charge transportation [102]. Earlier work also showed that trilayer-graphene/MoS_2_/n-Si solar cells achieved a PCE of 11.1%. MoS_2_ increases the built-in electric field due to an increased Fermi level between graphene and n-Si, which impacts the energy barrier between MoS_2_ and Si. Therefore, MoS_2_ is an electron-blocking layer and an HTL to the p-doped graphene [103]. 

## 3. Biomedical Applications:

### 3.1. Biomedical Applications of BN Nanostructures

#### 3.1.1. BNQDs: Properties and Applications

There is considerable interest in quantum dots (QDs) due to their applications in solar cells and biology [104]. These materials exhibit high surface-to-volume ratios, tunable bandgaps, physicochemical characteristics, and fluorescence. QDs are useful for bio-applications but contain heavy metals such as CdS [105], CdSe [106,107], and PbSe [108], posing risks to human health and ecosystems. Researchers are now focusing on metal-free QDs, which offer low cytotoxicity and are biocompatible. Metal-free QDs, such as carbon QDs (CQDs) [109,110], graphene QDs (GQDs) [111], carbon nitride QDs (CN-QDs) [112], and boron nitride QDs (BNQDs) [113], are being developed rapidly.

BNQDs are nanoparticles of hexagonal boron nitride (h-BN) sheets. Due to their metal-free, nontoxic nature, they have received increasing attention as members of the vdW materials family. BNQDs are <10 nm and consist of a few layers of a h-BN network. These materials are used for sensing, photocatalysis, chemotherapy, bioimaging, and detecting metal ions [114] (see GA). 

Several factors are involved in the fluorescence of BNQDs, including quantum confinement and structural defects. Quantum confinement allows size-dependent bandgap modulation, while defects introduce states within the bandgap for electrons and holes, facilitating defect-related emission when these carriers recombine. Additionally, the surface states of BNQDs, which can be modified through functionalization, also contribute to their overall fluorescent properties by providing additional pathways for electron-hole recombination. Due to these combined effects, BNQDs exhibit robust and adjustable fluorescence, making them suitable for bioimaging and other photonics applications [115]. 

#### 3.1.2. Photoluminescence in BNQDs

The mechanism behind BNQD photoluminescence is complex and subject to ongoing research. Their size and structural defects partially influence the luminescence of BNQDs. It has also been demonstrated that BNQDs with different sizes exhibit different luminescence characteristics, as demonstrated by Liu et al. [116], who synthesized BNQDs in different organic solvents with variable luminescence properties. The BNQDs produced using *N*-methyl-2-pyrrolidone (NMP) emitted a strong green fluorescence (501 nm), whereas those created using dimethylformamide (DMF) and ethanol emitted a blue fluorescence (436 nm and 420 nm, respectively). Different solvent polarities affect the interactions with boron nitride nanosheets, with higher-polarity solvents producing smaller BNQDs, resulting in a more pronounced red shift in fluorescence. The authors did not explain why smaller BNQDs produced by NMP emit fluorescence at a longer wavelength.

BNQDs exhibit luminescent behavior that is also affected by defects, doping, or vacancies within their structure. These can include carbon-substituted nitrogen vacancies, oxygen-deficient boron sites, and carbene structures along zigzag edges [114]. Using boron centers, Katzir et al. examined the effects of boron centers on luminescence, finding that such centers create trapping levels that, when interacted with by ionizing radiation, can produce blue photoluminescence. By altering the pH of the solvent, these defects and edges are more prevalent, resulting in altered photoluminescence of the BNQDs. According to Li et al., varying pH from acidic to alkaline conditions alters the intensity and wavelength of photoluminescence, suggesting that pH can control luminescence properties [117].

The current generation of BNQDs emit primarily blue to green light, which is a narrower spectrum than that of graphene quantum dots (GQDs), which exhibit a wider spectrum of visible light. Future research will focus on increasing the tunability of BNQDs’ photoluminescence to match the versatility of GQDs. BNQDs can be doped with various elements, such as halogens or chalcogens, to improve their photoluminescence range and enhance their utility across multiple applications [114]. 

#### 3.1.3. Biocompatibility and Cytotoxicity of BNQDs

Nanomaterials for biomedical applications must be biocompatible and of low toxicity. BNQDs composed of benign elements boron (B) and nitrogen (N) are superior in these regards compared to conventional quantum dots (QDs) such as CdSe and PbS, which may release toxic heavy metals into biological systems. The composition of BNQDs ensures an environmentally friendly and non-toxic profile, making them particularly appealing for biological analyses, drug delivery systems, cellular imaging, and diagnostic procedures [114]. 

Many studies have been conducted on the biocompatibility and cytotoxicity of BNQDs. The luminous cell imaging was examined using a confocal microscope in FITC (fluorescein isothiocyanate) mode by Lin et al. [118]. A few mammalian cell lines (MDCKII cells) were incubated in BNQDs for 24 h, and some nuclei were stained with DAPI (4′,6-diamidino-2-phenylindole). Examining these stained cells showed that BNQDs exhibit a brighter luminescence than DAPI. Based on the results of the cytotoxicity experiment, the authors concluded that low doses of BNQDs (0–40 g/mL) are not toxic to mammalian cells even after prolonged exposure for 24 to 48 h. BNQDs diffused through the hydrophobic lipid bilayer of the animal cell line. In contrast, Xue et al. [119] demonstrated that BN/BCNO QDs are biocompatible and less cytotoxic in RAW264.7 cells. The performance of cell imaging was also examined by incubating HeLa cells in BN/BCNO QDs for four hours. It was found that these QDs were taken up by HeLa cells through endocytosis during incubation. Furthermore, it was concluded that these QDs can penetrate through the cells without reaching the nucleus. Therefore, there is very little chance of genetic disruption. These BNQDs possess a bright fluorescence, making them attractive candidates for biological applications.

In a study by Lei and colleagues in 2015 [120], BNQDs were used to label He–La cells, and after a 7 h incubation, the cells exhibited distinct fluorescence when viewed with confocal microscopy. This indicated that BNQDs can penetrate cell membranes without affecting the cell’s genetic material, affirming their low toxicity. Additionally, the overlap of BNQD fluorescence with the Lyso Tracker red fluorescent probe suggests the entry of BNQDs into cells might be through endocytosis, localizing within the cytoplasm rather than the nucleus. Further research by the Allwood group demonstrated the effectiveness of BNQDs in staining MDCKII cells alongside DAPI, a nuclear stain, revealing BNQDs’ potential in producing high-contrast images for biological research. The accompanying confocal microscopy images provided clear evidence of the distinct luminescence of BNQDs. In 2019, HeLa cells were incubated with BNQDs and bright blue fluorescence was observed, mainly in the cytoplasm of the cell, with a confocal microscope. This shows BNQDs were internalized in the cells and proves the potential application of BNQDs in bioimaging probes [121]. 

#### 3.1.4. Electrochemical Luminescence in BNQDs

Electrochemical luminescence (ECL) is a method of converting electrical energy into radiant energy. The electroactive species will be transformed into an emissible fluorescent signal by applying the appropriate voltage to the electrode during the ECL procedure [122]. In addition to their chemical inertness, superior biocompatibility, and straightforward labeling, BNQDs are a viable electroluminescent material. A number of strategies have been developed in recent years to enhance the performance of BNQDs in addition to investigating their ECL characteristics. The surface states determine whether they exhibit ECL properties, and the mechanisms involved in the generation of ECL are different from those involved in photoluminescence. Various forms of doping and modification of BNQDs have been explored in order to improve their ECL performance [123]. 

To analyze the application of BNQDs in immune assays and biosensors, Xing et al. prepared an ECL sensor using Nafion and the cation of Tris(bipyridine)ruthenium(II) chloride [Ru(bpy)_3_^2+^] and dopamine (DA) [124]. Since Ru(bpy)_3_^2+^ has high ion exchange efficiency, this mixed electrode is believed to act as a performance booster. During this ECL reaction, DA reacts with other reagents to form o-benzoquinone (BQ), which is responsible for the quenching effect. On addition of BNQDs, the ECL emission from the electrode mixture increased abruptly, as high as 400 fold, as BNQDs enhanced the fluorescence. As shown in Figure 10a, the ECL intensity decreased as more DA was added to this mixture, and an irreversible oxidation peak was observed. In the figure, the ECL quenched efficiency [denoted as Y = (I_0_ − I)/I_0_, where I_0_ and I represent the ECL intensity in the absence and presence of DA, respectively] was in a linear relationship with the natural logarithm of the DA. The linear regression equation was Y = 0.23129 + 0.0903 log (Figure 10b). Furthermore, several small biomolecules like ascorbic acid (AA), glucose, L-cysteine (L-Cys), and lactic acid were chosen as the interference substance, and these biomolecules showed a slight change in ECL intensity. In addition, a human serum was used as a real sample to analyze the recovery. The recovery analysis suggested the use of a designed sensor for the detection of DA in a real serum sample [124]. 

In 2020, Kong et al. constructed a novel “on-off-on” fluorescence switch sensor to detect Fe^3+^ and ascorbic acid (AA) synchronously [125]. AA showed some effect in fluorescent intensity in BNQDs even without Fe^3+^. When Fe^3+^ was added to BNQDs, the quenching effect was observed, supposedly due to the inner filter effect (IFE). IFE occurs when the emission or excitation spectrum of the fluorophore overlaps with the absorption spectrum of the quencher. Later, the addition of AA in the BNQDs and Fe^3+^ mixture restored the fluorescence due to oxidation–reduction between Fe^3+^ and AA. The Fe^3+^ level was detected in Sha Lake water and human serum. This suggested the feasibility of the method and the potential biomedical application. Furthermore, to assess the selectivity of BNQDs with Fe^3+^ for AA, the effects of various types of interfering substances were studied, such as glutathione, histidine, glucose, oxalic acid, cysteine, riboflavin, glycine, creatinine, arginine, citric acid, dopamine with the same concentration [125]. 

#### 3.1.5. BNNTs: High-Brightness Fluorophores

BNNTs are electrically insulating and optically transparent due to the large bandgap ~6 eV [20,126,127,128,129]. The nanotube surfaces are thoroughly saturated with covalent bonds, making BNNTs chemically inert for biological applications, much like CNTs. In this regard, Ciofani et al. are among the researchers who started to study the biomedical applications of BNNTs [130,131], replicating biomedical applications of CNTs. In this subsection, we will review some of these emerging applications of BNNTs based on their unique properties not found in other nanomaterials.

Yap et al. have created high-brightness fluorophores (HBFs) based on BNNTs [12,13,132]. The electrically insulating nature of BNNTs maintains the quantum yield of dye molecules attached to BNNTs. The authors managed to enhance the molar extinction coefficients up to 1000× by organizing arrays of dye molecules on each BNNT. The molar extinction coefficients are scalable between 10^8^ to 10^11^ M^−1^cm^−1^ [132,133,134,135]. The design of such HBFs is shown in Figure 11. For example, they conjugated one FITC molecule (fluorescein isothiocyanate) with one DSPE-PEG5K-NH_2_ linker {amine-functionalized (1,2-distearoyl-sn-glycero-3-phosphoethanolamine-*N*-[(polyethylene glycol)_n_]}, to form one unit of dye-linker (Figure 11a). These dye-linkers are then non-covalently functionalized on BNNTs [132]. The appearance of a high-brightness dye is schematically illustrated in Figure 11b. Such BNNTs functionalized with DSPE-PEG are well dispersed in water for biomedical applications [136]. 

These high-brightness dyes can be conjugated with other biomolecules, such as antibodies, to formulate HBFs, for example for detecting antigens on peripheral blood mononuclear cells (PBMC) by FCM. Preliminary FCM data demonstrating the high fluorescent brightness presented in conferences are promising [132,133,134,135]. The insulating nature of BNNTs has enabled the formation of such HBFs. HBFs could not be constructed using CNTs due to the metallic nature of the carbon nanomaterials. Many researchers have reported that dye molecules labeled on CNTs would quench by 20–80% due to electron transfer [137,138,139,140,141].

### 3.2. Biomedical Applications of MoS_2_ Nanostructures

#### 3.2.1. Drug Release by External Stimuli

MoS_2_ has received considerable attention for biomedical applications such as tissue engineering, biosensing, imaging, and drug delivery. For example, MoS_2_ can be an effective drug carrier because of its large surface area and capacity to functionalize with biomolecules. Drugs can be added to the MoS_2_ nanosheets to protect the drug from early deterioration and improve its stability during circulation. The adjustable drug release rate also enables regulated and targeted drug administration, minimizing adverse effects and enhancing therapeutic efficacy. External stimuli like pH, light, or temperature changes initiate this behavior.

Khodabakhshi et al. studied the development of magnetic MoS_2_ nanocarriers based on thermosensitive dendrimers [142]. They synthesized thermosensitive dendrimers based on magnetic MoS_2_ as the nanocarriers (M-MoS_2_/MMA-G 5/L-A/PVE), as illustrated in Figure 12.

In the investigation (Figure 12), the adsorption and release of cisplatin were examined for potential therapy. The highest adsorption was discovered at pH 7, with a contact period of 10 min. The drug release capabilities were tested at various pH levels that replicated the environments of cancer cells (pH 5.6) and healthy body cells (pH 7.4). Cisplatin is released more effectively from the M-MoS_2_/MMA-G 5/L-A/PVE smart nanocarrier than from the nanocarrier without PVE. Here, PVE (polyvinyl ether) works as a thermos-sensitive polymer. The release of cisplatin from M-MoS_2_/MMA-G 5/L-A/PVE at pH 5.6 and 50 °C is highest, with a burst in the initial 1h to 48%, followed by a steady release up to 87% after 6h. The release from the nanocarriers without PVE is much lower in the same condition. The release is also low at pH 5.6 and 37 °C. On the other hand, the release at pH 7.4 and 50 °C reached 43% and 80% in the first 30 min and after 6 h. This result suggests that the smart nanocarrier was temperature-dependent due to using PVE. In addition, drug release from the smart nanocarrier can also be triggered by near-infrared (NIR). According to the NIR investigation, cisplatin was released at pH 7.4 and 5.6, respectively, releasing 86% and 92% of the drug.

The cytotoxicity of MoS_2_ is debated [143,144]. To address this issue, Zhang et al. developed a unique method for surface-modifying MoS_2_ nanoparticles (NPs) with cationic hydroxyethyl cellulose (JR400), resulting in JR400-MoS_2_ NPs that respond to near-infrared (NIR) laser irradiation for transdermal drug administration [145]. The JR400-MoS_2_ NPs were synthesized by hydrothermal process and had a flower-like shape with an average diameter of ~355 nm, as shown in Figure 13. JR400-MoS_2_ NPs are distinguished by their ability to efficiently load a model drug, atenolol (ATE). It was shown that 1g of JR400-MoS_2_ NPs may load up to 3.6 g of ATE.

In vitro skin penetration experiments revealed that the NPs had a regulated release capability, enabling precise drug administration. Notably, when subjected to NIR stimulation, the JR400-MoS_2_ NPs achieved a 2.3-fold increase in ATE skin penetration compared to passive release. The biocompatibility and safety profile of any medication delivery technology are critical. JR400-MoS_2_ NPs did not induce skin irritation in vivo, highlighting their potential as attractive new candidates for transdermal drug delivery systems for small-molecule medicines. The newly developed strategy of employing JR400 surface-modified MoS_2_ NPs for transdermal drug delivery eliminates the toxicity problems associated with MoS_2_ nanotubes and unlocks their unique possibilities for medicinal applications [145].

#### 3.2.2. Label-Free Virus Sensors

Dong et al. reported colorimetric sensors for detecting label-free avian influenza A (H7N9) viral gene sequences [146]. The sensor is assembled from gold and platinum core-shell bimetallic-nanoparticle-decorated molybdenum disulfide (MoS_2_Au@Pt) nanocomposites. As nanoenzymes, MoS_2_Au@Pt nanocomposites play an important role in this sensing platform, as shown in Figure 14. They have inherent peroxidase-mimicking activity, allowing them to catalyze the hydrogen peroxide (H_2_O_2_) conversion of 3,3′,5,5′-tetramethylbenzidine (TMB). This catalytic activity acts as a signal amplification mechanism, increasing detection sensitivity. The differential affinities of the MoS_2_Au@Pt nanocomposites towards single-stranded (ss) and double-stranded (ds) DNA, as well as the target-triggered catalyzed hairpin assembly (CHA) process, enable the sensor’s specificity and quantification capabilities. The presence of the target H7N9 gene sequence starts the CHA reaction, resulting in a certain DNA structure.

As a result, MoS_2_Au@Pt nanocomposites preferentially bind to the DNA structure, further boosting the colorimetric signal and allowing for both qualitative and quantitative detection of the H_7_N_9_ virus. One of the most notable advantages of this sensor is its simplicity. The adsorption spectra of the sensor for H7N9 analysis show an increase in peak intensity at 652 nm with higher H7N9 concentrations (from 10 pM to 50 nM). This indicated that the added H7N9 triggered the release of HP1 and HP2 from the MoS_2_Au@Pt nanocomposites’ surface, leading to the recovery of their catalytic activity. The solution color changed from light blue to navy blue, confirming the successful analysis of H7N9. The sensor’s detection limit was 2.8 pM, demonstrating higher sensitivity than fluorescence and electrochemical methods.

#### 3.2.3. Piezopotential Neuron Stimulation

Fan et al. studied piezoelectric MoS_2_ nanosheets (MoS_2_ NS) to transform ultrasonic energy into a localized electrical field remotely [147]. Such piezopotential stimulation could perturb the membrane potential of neurons to allow an influx of Ca^2+^ ions into the SH-SY5Y cells as illustrated in Figure 15a. The acoustic pressure, number of ultrasonic cycles, and MoS_2_ NS concentration determine the stimulated cell fraction. The potential of ultrasonic stimulation based on MoS_2_ NS was further substantiated by in vivo evidence. A measure of neural activity called c-Fos was three times more prevalent in neurons near the MoS_2_ NS than it was in neurons farther away from the nanosheets. This effective stimulation of the neurons around the MoS_2_ NS revealed a technique for selectively altering particular brain circuits with great spatial accuracy. The new approach represents a substantial improvement in brain stimulation and has considerable potential for future neuroscience research and therapeutic applications.

According to in vitro tests, using ultrasound to treat the cells around MoS_2_ NS only required one pulse of 2 MHz at specific settings (400 kPa, 1,000,000 cycles, and 500 ms pulse length). Surprisingly, this led to substantial responses in 37.9% of the cells, without resulting in any observable cellular damage. Ionomycin is used as the inducer, capable of activating Ca^2+^ fluorescence signals. Figure 15b shows the time-lapse Ca^2+^ images of SH-SY5Y cells at different stimulation conditions, suggesting the successful activation of intracellular Ca^2+^ dynamics. The difference between initial fluorescence intensity at the resting state and after stimulation/initial fluorescence intensity at the resting state, Δ*F*/*F*_0_ peak = 0.2 ± 0.1, as shown in Figure 15c.

#### 3.2.4. Photothermal–Chemo Therapy

In the field of tumor treatment, the combination of several treatment modalities has attracted substantial attention to improving therapeutic outcomes. Liu et al. used MoS_2_ as the core material with photothermal conversion capabilities, as the core material for photothermal–chemo therapy [148]. Manganese dioxide (MnO_2_) was loaded onto the surface of MoS_2_ to produce a mesoporous core-shell structure, taking advantage of MnO_2_’s response to the tumor microenvironment. The anticancer agent Adriamycin (DOX) was injected into the nanocomposite’s mesoporous pores. The surface of MnO_2_ was modified with mPEG-NH_2_ to improve its water solubility and stability, culminating in the creation of the mixed anticancer nanocomposite MoS_2_@DOX/MnO_2_-PEG (MDMP).

The diameter of the nanocomposite MDMP was around 236 nm, and its photothermal conversion efficiency was 33.7%. DOX loading and release rates were notable at 13% and 65%, respectively, demonstrating efficient drug loading and regulated release capabilities. MDMP displayed remarkable anticancer efficacy in both in vivo and in vitro experiments. Figure 16a illustrates how the tumor volume changed throughout treatment. Compared to the blank group, the administration group showed a significant therapeutic effect, and the MDMP + NIR group exhibited even better results than the MDMP group alone. After 14 days of treatment, the tumors were dissected from the mice, photographed, and weighed (Figure 16b). The tumor photos demonstrated that the combined therapy had the most favorable effect. Furthermore, the tumor weight in different treatment groups varied (Figure 16c). The blank group had the heaviest tumors, while the MDMP + NIR group displayed the smallest tumor weight. Throughout the 14-day treatment period, the body weight of the mice in the MDMP, MDMP + NIR, and blank groups remained unchanged, as indicated in Figure 16d. This result suggests that the treatments were well tolerated by the animals and did not cause significant fluctuations in their body weight. The combination therapy yielded a relatively low tumor cell viability rate of 11.8%. This finding underlines the MDMP nanocomposite’s tremendous potential for chemo–photothermal combination anticancer treatment. The unique combination of MoS_2_ as a photothermal agent, MnO_2_ as a tumor microenvironment-sensitive material, and DOX as a chemotherapeutic medication demonstrate the adaptability and efficacy of this nanocomposite in treating tumors via numerous routes. The effective in vitro and in vivo experiments highlight MoS_2_ nanocomposites’ significant promise in developing tumor treatment techniques. The chemo–photothermal combination therapy method can improve treatment efficacy while minimizing adverse effects, making MDMP an appealing option for future investigation and potential translation into clinical cancer therapeutic applications [148]. 

## 4. Environmental Applications

### 4.1. Environmental Applications of BN Nanostructures

#### 4.1.1. Sensors and Gas Capture

Environmental monitoring has become essential due to increasing pollution and contamination of the air and water. The main aspect of removing contamination from the environment is detecting it. It was reported that BNNTs and BN sheets can be used to capture one the most dangerous compounds, (2,3,7,8) Tetrachlorodibenzo-p-dioxin (TCDD). TCDD is a colorless dioxin-related compound with no distinguishable odor and is usually formed in the burning processes of organic compounds. Density functional theory (DFT) suggests that the TCDD compound adsorbs parallelly on BNNTs and is stronger than CNTs. The adsorption of TCDD is also stronger on pristine BNNT than on the defective ones [149]. The adsorption of gas molecules on BN sheets is also studied. Guo et al. studied the adsorption of gases (CO_2_, H_2_, CH_4_, N_2_, CO, and H_2_O) with and without an external electric field using DFT [139]. The calculated binding energy without an external electric field is 0.44 eV for CO_2_, significantly higher than for other gases (0.13 to 0.26 eV). The adsorption is determined by van der Waals interaction. The external electric field escalated these adsorptions, and the binding energy of the molecules increased. Therefore, h-BN sheets could be used as the CO_2_ capture materials in the mixture of other gases such as H_2_, CH_4_, N_2_, CO, and H_2_O. Another study focuses on the theoretical analysis (using DFT) of armchair (5,5) BNNTs for HX (where X = F, Cl, Br) adsorption on the surface of BNNTs in the gas phase. HX molecules adsorb on the BNNT surface exothermally with favorable adsorption energy values. The study concludes that BNNTs have the potential to be a sensor detecting HX molecules, with HF exhibiting the strongest adsorption [150]. In another study, SnO_2_ nanoparticles were loaded onto BNNTs using RF sputtering to create a SnO_2_-BNNTs-based gas sensor for very sensitive detection of NO_2_ which has improved sensor response as compared to SnO_2_ sensors. At a low operating temperature of ~100 °C, the sensor exhibits a five-fold increase in reaction to NO_2_, with a significant response value of 119.6 (24.6 without BNNTs). The modulation of the space charge depletion layer in the p-n heterojunctions of SnO_2_-BNNTs and the current fluctuation in conjunction with NO_2_ gas are both responsible for the increased sensor responsiveness. At low operation temperatures (25–300 °C), BNNTs offer charge carriers conducting channels, which improves responses [151]. On the other hand, the adsorption of noble gases was studied on the BN sheets, and using the particle swarm optimization technique, the global optimization of He, Ne, Ar, and Kr was attained [152]. Zhu et al. reported the optimum catalytic effect of Pt nanoparticles on h-BN nanosheets [153]. Experiments suggest that strong interaction exists between Pt and the N and B vacancies of h-BN. The nanosheets act as a Lewis base to donate electrons to Pt, making Pt favor absorbing O_2_, alleviating CO poisoning, and promoting the catalytic effects.

#### 4.1.2. Desalination and Water Treatment Membranes

As the demand for freshwater increases with the increase in population, various research has been carried out to remove salt from seawater. Multi-stage distillation nuclear desalination techniques which needed high operation pressures were commonly used. The ion concentration polarization (ICP) process achieved high power efficiency comparable to state-of-the-art reverse osmosis (RO) plants. The process removed 99.9% of salt from seawater in a single-step operation, recovering 50% as desalted water. A zero discharge desalination technique was also used [154,155,156,157].

Nanotechnology has proved to be an efficient way for seawater desalination. A molecular dynamics simulation was performed to investigate the wetting behavior of BNNT (5,5) and CNT (5,5). It was found that water molecules permeate through a BNNT (5,5), while a CNT (5,5) does not conduct water. The relatively strong van der Waals interaction between the N atoms of BNNTs and water molecules plays a key role in the continuous wetting behavior of the BNNT [158]. Liang et al. performed molecular dynamics simulation on water filtering and salt rejection of BNNTs [159]. Water molecules can travel through BNNT (5,5) and BNNT (6,6) one by one. With the increased diameter, more than one water molecule can pass through BNNT (7,7) and BNNT (8,8) simultaneously. Different BNNT chiralities have different water molecule densities and orientations. With increased diameters and pressures, BNNTs can pass more water through them as shown in Figure 17. The salt rejection of BNNT (5,5) and BNNT (6,6) is 100%, but the water flux is up to ~47–61 molecules/ns at a pressure of 500 MPa. Excellent desalination capabilities are displayed by BNNT (7,7), which has higher water permeation and salt rejection rates (>98%) even at lower pressures of 300–400 MPa. The salt rejection of BNNT (8,8) is <80% in most cases. In BNNT (7,7), the selectivity of water over sodium ions is thought to be as high as 3071:1 [159]. Arrays of BNNT (7,7) membrane (10 cm^2^, with 1.5 × 10^13^ pores per cm^2^) will produce 98L freshwater per day per MPa under 100 MPa. It is two orders higher than the commercial reverse osmosis membrane.

Functionalized BNNT (8,8) arrays, BNNT (8,8)-COO^−^ and BNNT (8,8)-NH_3_^+^ were also constructed for desalination applications [160]. The free-energy profiles for water molecules and ions passing through functionalized BNNT (8,8) were calculated. The free-energy profile showed that water molecules have lower energy barriers compared to ions which indicated that water molecules can pass through more easily and ions tend to stay out of the nanotube since the transfer behavior of ions in functionalized BNNT (8,8) is not favorable thermodynamically and dynamically which in turn improves the salt rejection and desalination performance [160]. 

BNNT membranes have an advantage over CNTs for rejecting negatively charged particles from water [161]. The flow resistance of BNNTs was studied by combining experimental data and simulation. The nature of the nanotube–membrane interface and the presence of partial charge discontinuities impact the end resistance, and in BNNT flow resistance, electrostatic interactions play a vital role and the accuracy of molecular dynamics simulations depends on the selection of partial charges. For small-diameter membranes like BNNT, more research is required to improve the definition of partial charges. Chen et al. have studied the impact of -NH_3_^+^ functional groups and their location on desalination performance (Figure 18) [162]. They found that electrostatic interaction has a greater effect on desalination performance than a steric hindrance. The behavior of water molecules is affected by the presence of electric charge which leads to a decrease in the water flux rate as the molecules adjust their configuration while entering the nanotube. The salt rejection rate is increased due to electrostatic interaction, as it blocks the Cl^−^ ions near the nanotube entrance and decreases the water flux rate.

#### 4.1.3. Corrosion Protection

Corrosion causes a change in appearance and decomposition of materials in stainless steel. Since many industrial fields use stainless steel, corrosion has become a very constant and challenging problem [163]. Two-dimensional BN coatings can provide extra barrier property against corrosion and oxidation under aggressive media [164].

Xuemei et al. described a method to directly grow large-area continuous h-BN nanofilms of thickness ~200 nm on grade 304 stainless steel (ss304) surfaces to provide a protective coating for applications in corrosive environments using the RF magnetron sputtering method. The films have excellent resistance to oxidation of stainless steel at high temperatures (~600 °C) for 30 min [163]. Another group studied the BN coating on ss304 using the plasma vapor deposition (PVD) method. It analyzed the chemical composition and surface morphology of BN coating using ATR-FTIR, Raman spectroscopy, XRD, and FESEM in 1M H_2_SO_4_ solution [164]. BN-coated stainless steel displayed a positive shift in open circuit potential, which indicates the BN film’s effective corrosion protection ability and the metal’s passive state with a protection efficiency of 84%. Combining BN and Ti_3_C_2_T_x_ (one type of transition metal carbides, nitrides or carbonitrides, MXenes) improves corrosion protection [165]. It was suggested that a Ti_3_C_2_T_x_/BN/epoxy coating demonstrates excellent corrosion protection properties and compactness of the coating, which slows down the diffusion of corrosive media. The Ti_3_C_2_T_X_ MXene with great electrical conductivity accelerates substrate corrosion by forming a conductive network. The insulating BN nanosheetsare repulsive to MXene reduce the formation of conductive network, gives rise to outstanding corrosion protection exhibited by the Ti_3_C_2_T_x_/BN/epoxy coating.

### 4.2. Environmental Applications MoS_2_ Nanostructures

#### 4.2.1. Photocatalyst

In one study, photocatalytic activity was studied using different catalysts to decolorize and degrade methylene orange under simulated solar light. The addition of MoS_2_ to g-C_3_N_4_ significantly improved photocatalytic efficiency, with the optimal ratio being 3% due to increased surface area and charge transfer [166]. The composite C_3_N_4_-MoS_2_ photocatalysts were prepared by a hydrothermal method. Another study found that the MoS_2_@ZnO composite exhibited stronger adsorption capabilities in the dark-enhanced photocatalytic activities compared to pure MoS_2_ nanoflowers [167]. TiO_2_-MoS_2_ nano-heterojunctions were also studied and showed promise for applications in photocatalysis [168]. Another study explored using TiO_2_ nanoparticles for photocatalytic organic waste degradation in water. To overcome the limits of traditional TiO_2_, various modification methods were explored by coupling TiO_2_ with different semiconductor materials. MoS_2_ matches the solar spectrum yet had challenges with electron transfer to TiO_2_. In the study, TiO_2_/MoS_2_ photocatalyst was created with bulk MoS_2_ as a direct photosensitizer which significantly enhanced the photocatalytic activity under visible light by facilitating efficient electron transfer from nano-MoS_2_ to TiO_2_, resulting in the degradation of organic pollutants like methyl orange [169]. 

On the other hand, MoS_2_ are combined with graphene oxide (GO) and reduced GO (rGO) to degrade methylene blue in visible light [170]. The authors studied the structural and morphological changes in GO/MoS_2_ and rGO/MoS_2_ composites. XRD measurements revealed that GO/MoS_2_ maintained MoS_2_’s crystalline structure after sonication, while GO underwent fragmentation and the mean thickness of MoS_2_ decreased moderately after sonication. After thermal treatment, rGO/MoS_2_ partially restored the graphite phase. The optical properties indicated a reduction in oxygen functional groups in rGO/MoS_2_, as seen in UV-vis and DRIFT spectra. Photodegradation tests demonstrated that rGO/MoS_2_ outperformed pure MoS_2_, rGO, and TiO_2_ in degrading methylene blue in visible light [170]. MoS_2_ nanocomposites were studied for photocatalytic degradation of methyl red dye [171]. In this study, a ZnO-MoS_2_ composite was prepared by decorating ultrafine ZnO nanoparticles onto exfoliated MoS_2_ nanosheets which displayed a remarkable photocatalytic efficiency by achieving 89% methyl red dye decolorization in 60 min under solar light irradiation, surpassing the performance of pure ZnO nanoparticles (69%) and exfoliated MoS_2_ nanosheets (55%) [171]. Li et al. studied a 3D MoS_2_@TiO_2_@poly (methyl methacrylate) nanocomposite which gave enhanced photocatalytic performance [172]. Panchal et al. studied engineered MoS_2_ nanostructures for improved photocatalytic application in water treatment [173]. 

Apparently, MoS_2_ alone is not responsible for the enhanced photocatalytic effects reported in the above-mentioned publications. The combined properties of the reported hybridized materials show interesting synergistic effects for applications, such as facilitating efficient electron transfer.

#### 4.2.2. MoS_2_-Based Sensors

MoS_2_-based sensors exhibit higher selectivity than CNT-based sensors [174]. Yan et al. synthesized ZnO-coated (8 nm) MoS_2_ nanosheets (500 nm) by a two-step hydrothermal method and tested their gas-sensing performance towards ethanol [175]. It was noted that in comparison to ZnO-nanoparticles, ZnO-coated MoS_2_ showed better sensing performance towards ethanol. In another study, SiO_2_ nanorods (NRs) encapsulated by MoS_2_ were used to enhance the sensing properties of NO_2_ gas [176]. The sensors exhibited high sensitivity, with a detection limit of approximately 2.4 ppb at room temperature. Compared to MoS_2_ on a flat SiO_2_ substrate, the encapsulated NRs showed a 90-fold increase in gas response when exposed to 50 ppm of NO_2_. At an operating temperature of 100 °C, the sensors demonstrated reliable recovery and a detection limit of around 8.84 ppb. The fabrication technique enabled the exposure of edge sites of MoS_2_, which played a crucial role in gas adsorption [176]. 

A novel MoS_2_/VS_2_ heterostructure was synthesized by vertically growing MoS_2_ nanosheets on a porous VS_2_ scaffold. A quartz crystal microbalance (QCM) sensor based on this heterojunction showed high sensitivity and selectivity towards ammonia. The computational simulations confirmed its superior adsorption energy for ammonia compared to individual MoS_2_ of VS_2_ structures [177]. Another research group developed a porous platform, h-MoS_2_/Pt, with improved sensing speed, selectivity, and stability [178]. They focused on designing ultrathin MoS_2_ shells using Pickering emulsification. The resulting h-MoS_2_/Pt sensors exhibited excellent hydrogen-sensing capabilities at room temperature in the presence of air. The outstanding performance was attributed to the high surface area of the ultrathin MoS_2_ shell and the spillover effect of Pt nanoparticles in sensors [178]. A MoS_2_/WO_3_ nanocomposite synthesized through a top-down approach showed superior performance as an ammonia sensor at 200 °C [179]. The composite exhibited increased specific surface area and formed heterojunctions between MoS_2_ and WO_3_ [179]. In another study, the gas-sensing performance of hybrid nanostructures consisting of metal oxide and 2D transition metal dichalcogenides (Co_3_/MoS_2_) was studied [180]. By varying the precursor concentration and operating temperature, the sensors exhibited excellent sensitivity towards multiple gases, with the sensors using a 25 mM precursor showing the highest response and stability. At the operating temperature of 250 °C, the sensor demonstrated exceptional efficiency in detecting NO_x_ gases [180]. Hingangavkar et al. reported that when a reducing gas was applied to the MoS_2_ nanosensor, the resistance of the sensor increased, and replacing the reducing gas with air caused the resistance to return to its initial state [181]. The increase in resistance of p-type MoS_2_ upon exposure to H_2_S is due to the electron-donating behavior of reducing gases and minority charge carriers, holes, decreases in concentration on the surface of MoS_2_ when gas molecules are absorbed, resulting in the decrease in conductivity due to depletion of majority charge carrier concentration caused by the presence of H_2_S gas [181]. In another study, a hybrid nanocomposite of MoS_2_ nanoflakes and two-dimensional (2D) Al_70_Co_10_Fe_5_Ni_10_Cu_5_ quasicrystal (QC) nanosheets (2D-QC) was synthesized. The composite showed improved conductivity during exposure of NO_2_ gas (Figure 19) [182]. 

## 5. Summary and Future Perspective

In summary, we have reviewed the most recent progress in using BN and MoS_2_ nanostructures in three areas: electronic and energy, biomedicine, and environmental applications. We have observed two major research trends: (1) Researchers are exploring heterogeneous materials by combining BN or MoS_2_ nanostructures with other materials for applications. (2) In addition to popular applications in electronics, energy, and biomedicine, researchers are actively pursuing environmental applications such as CO_2_ gas capture, water desalination, corrosion protection, photocatalytic decomposition of pollutants, and gas sensing. Future research should continue to explore these two major trends.

On top of these, researchers should start thinking about how to purify soils and water that are contaminated with nanomaterials due to the increased use of nanostructures in applications. As mentioned in the introduction (and Figure 1), contamination in air, soil, and water will eventually end in water bodies, rivers, lakes, and oceans. Potentially, nanomaterials may enter into our food chains, much like the case of microplastic. As discussed in the introduction, we developed a simple, universal, and effective method to remove nanomaterials from contaminated water [9,10]. Our approach involves emulsifying the contaminated water with organic liquids (for example, cooking oil). The interface between water and oil particles will introduce forces to trap nanomaterials. When water and oil separated after resting, all nanomaterials were trapped within the organic phase on top of the water.

## Figures and Tables

**Figure 1 micromachines-15-00349-f001:**
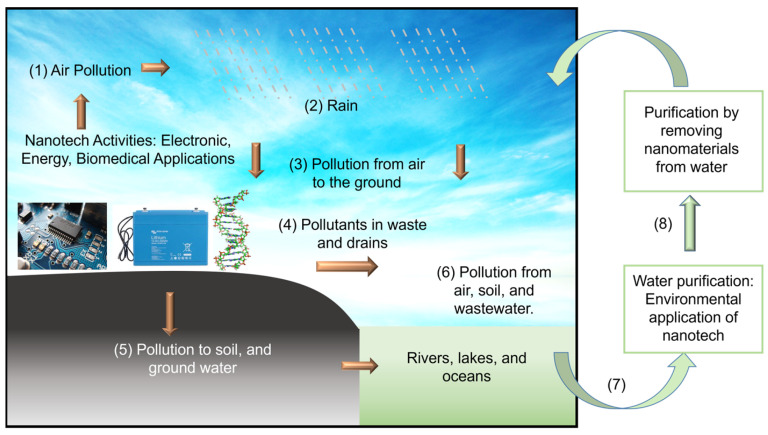
Schematic illustration of the importance of studying the environmental application of nanomaterials. (1) Air pollution from nanotech activities will eventually bring pollutants to soil, lakes, rivers, oceans, and groundwater via processes (2) to (6). The unintentional contamination of water by nanomaterials during environmental application is highly possible. Therefore, research to purify water by extracting nanomaterials from water (7 and 8) becomes critical [9,10]. Credit: Y. K. Yap.

**Figure 2 micromachines-15-00349-f002:**
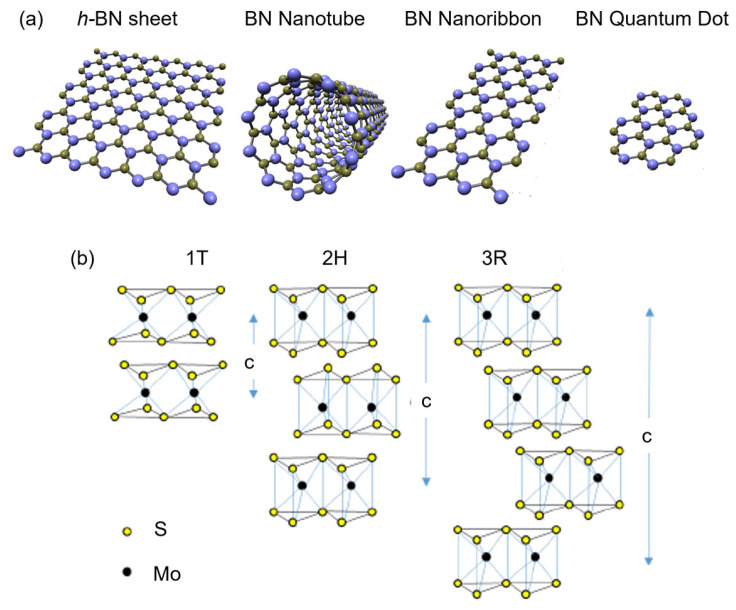
Schematic illustration of (**a**) BN and (**b**) MoS_2_ nanostructures. Credit: Y. K. Yap.

**Figure 3 micromachines-15-00349-f003:**
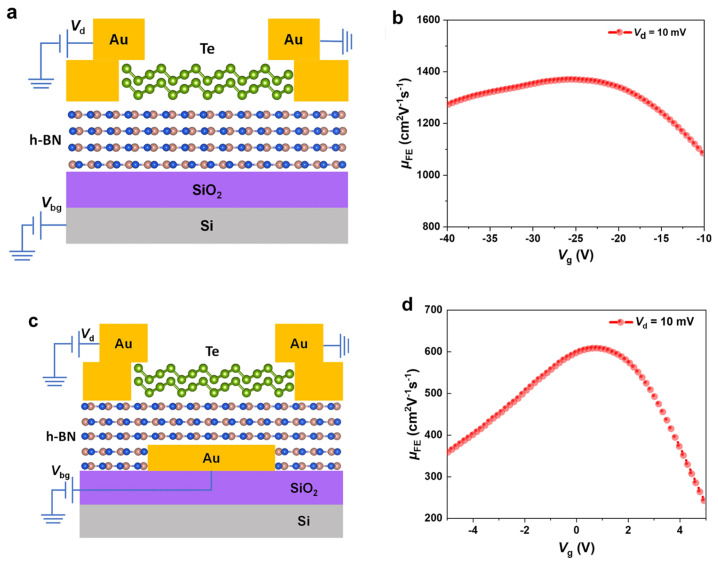
(**a**) Schematic illustration of Te-based FET with global bottom-gate structure on h-BN/SiO_2_/Si substrate in a cross-sectional view. (**b**) Field-effect mobility of Te transistor extracted from the transfer curves under the bias of V_d_ = 10 mV in the panel. (**c**) Schematic illustration of local bottom-gate Te FET by using h-BN as a dielectric layer in a cross-sectional view. (**d**) Field-effect mobility of Te transistor extracted from the transfer curve under the bias of V_d_ = 10 mV [30]. Reproduced with the permission of Springer (Copyright 2022).

**Figure 4 micromachines-15-00349-f004:**
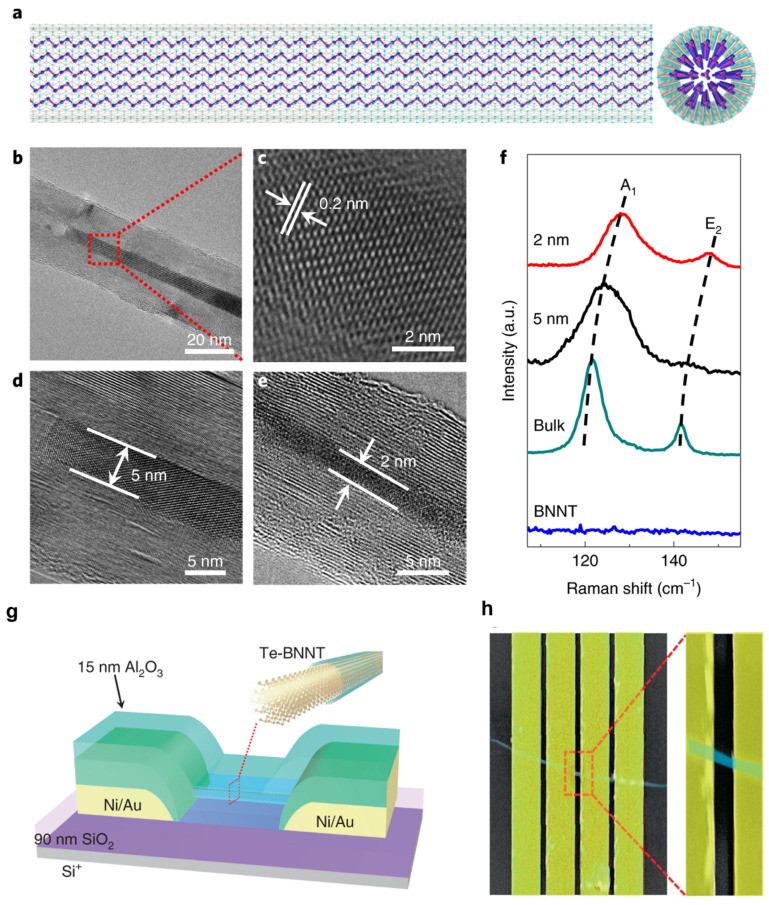
(**a**) Schematic illustration of a Te-BNNT. (**b**) TEM image of a 5 nm Te NW in a BNNT. (**c**) Enlarged HRTEM image of the region outlined in red in (**b**). (**d**,**e**) HRTEM images of BNNTs filled with 5 nm (**d**) and 2 nm (**e**) Te NWs. (**f**) Raman spectrum comparison of Te NWs in BNNTs with different diameters as indicated. (**g**) Schematic illustration of an individual Te-BNNT FET. (**h**) False-colored SEM image of a representative FET device before Al_2_O_3_ capping [25]. Reproduced with the permission of Springer Nature (Copyright 2020).

**Figure 5 micromachines-15-00349-f005:**
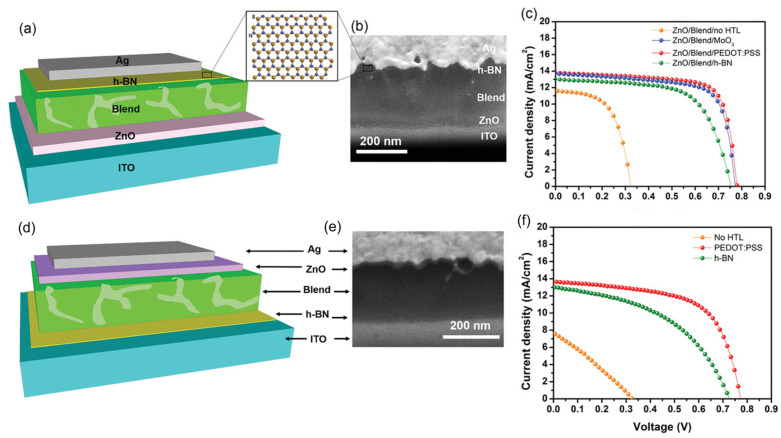
(**a**) Schematic and (**b**) cross-sectional SEM image of the Organic Photovoltaics with inverted device structure. (**c**) J–V curves without HTL and with MoO3, PEDOT: PSS, or h-BN. (**d**) Schematic and (**e**) cross-sectional SEM image of the OPV with conventional device structure and (**f**) J–V curves of the OPVs without HTL and with PEDOT: PSS or h-BN [38]. Reproduced with the permission of Wiley (Copyright 2021).

**Figure 6 micromachines-15-00349-f006:**
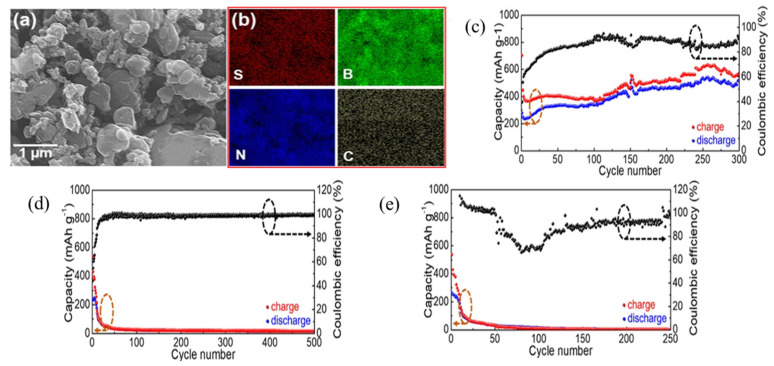
(**a**) SEM image. (**b**) EDX mapping of the BN/S/C sample. (**c**) Long-term repeated charge/discharge cycling measurements for BN/S/C at a current density of 100 mAg^−1^ within a potential window of 0.05–2.2 V vs. AlCl_4_^−^/Al (**d**) MoS_2_/S/C and (**e**) WS_2_/S/C [53]. Reproduced with the permission of Springer Nature (Copyright 2019).

**Figure 7 micromachines-15-00349-f007:**
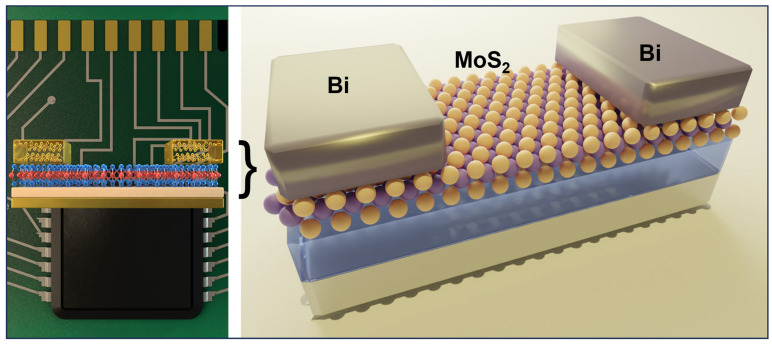
Illustration of an FET based on Bi/MoS_2_ contacts. The Bi (0001) plane is parallel to the plane of MoS_2_ [75]. (Credit: Massachusetts Institute of Technology).

**Figure 8 micromachines-15-00349-f008:**
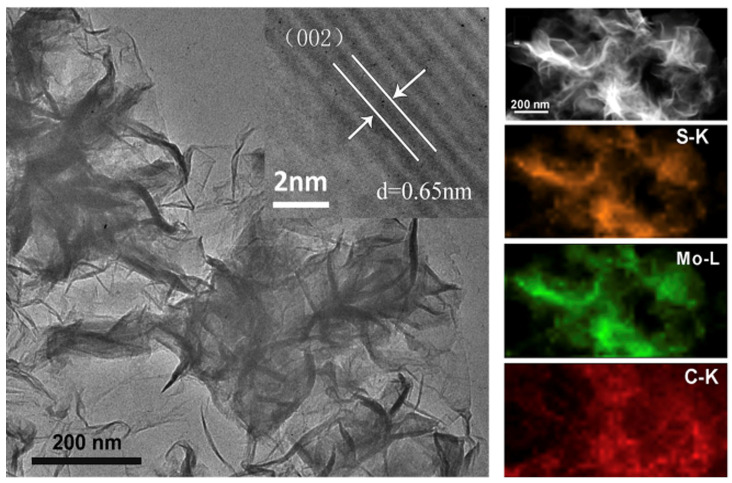
TEM (**left**) and HRTEM (**inset**) images of the flower-like 3DG/MoS_2_. TEM image (**upper right**) of 3DG/MoS_2_ and the corresponding EDX elemental mapping of S, Mo, and C [93]. Reproduced with the permission of Elsevier (Copyright 2016).

**Figure 9 micromachines-15-00349-f009:**
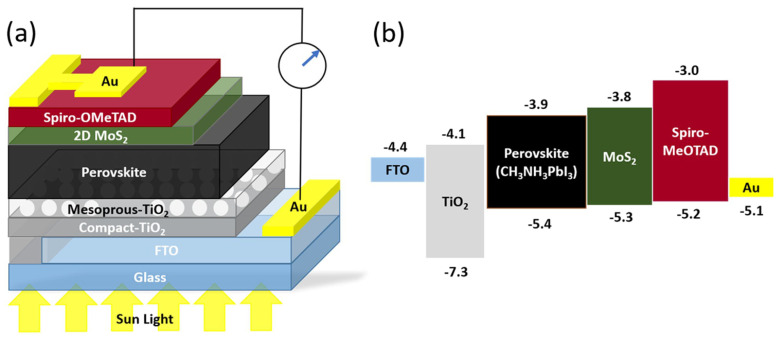
(**a**) Schematic of the perovskite solar cell with MoS_2_ nanoflakes. (**b**) Energy band diagram of the device shows the function of MoS_2_ as an additional HTL [102]. Reproduced with the permission of Springer Nature (Copyright 2020).

**Figure 10 micromachines-15-00349-f010:**
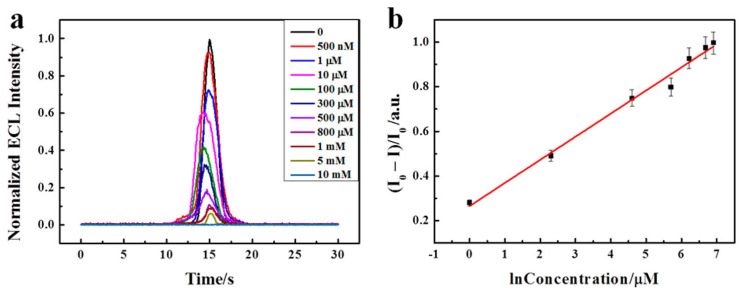
(**a**) Normalized ECL intensity of sensor with different concentrations of DA (0, 0.5, 1, 10, 100, 300, 500, 800, 1000, 5000, 10000 μM). (**b**) Linear relationship between (I_0_ − I)/I_0_ and the natural logarithm concentration of DA in the range of 1−1000 μM [124]. Reproduced with the permission of America Chemical Society (Copyright 2018).

**Figure 11 micromachines-15-00349-f011:**
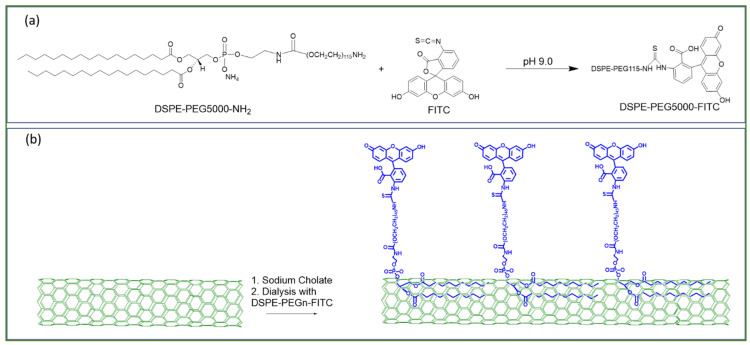
Schematic drawing of (**a**) a DSPE-PEG-NH linker with a FITC molecule to form a dye-linker. (**b**) Non-covalent functionalization scheme to conjugate dye-linkers on each BNNT [13]. Reproduced with the permission of Springer Nature (Copyright 2020).

**Figure 12 micromachines-15-00349-f012:**
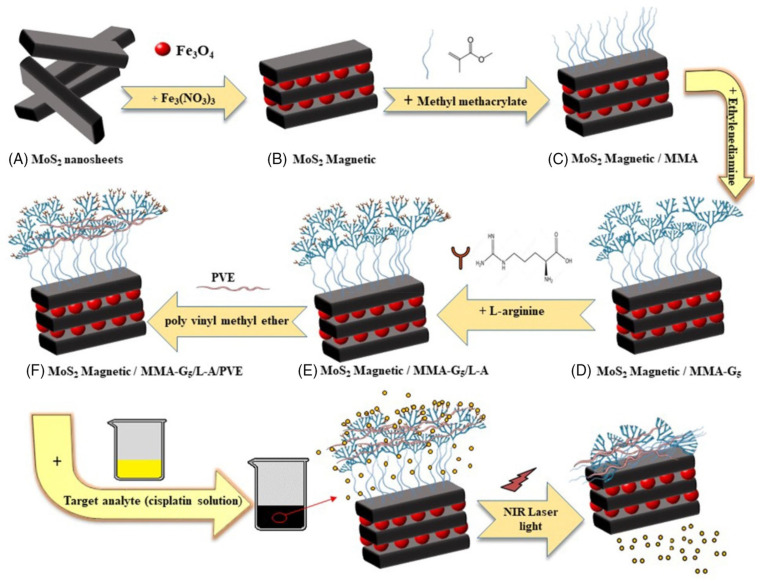
Schematic illustration of the nanocarriers (M-MoS_2_/MMA-G 5/L-A/PVE) [142]. Reproduced with the permission of Wiley (Copyright 2021).

**Figure 13 micromachines-15-00349-f013:**
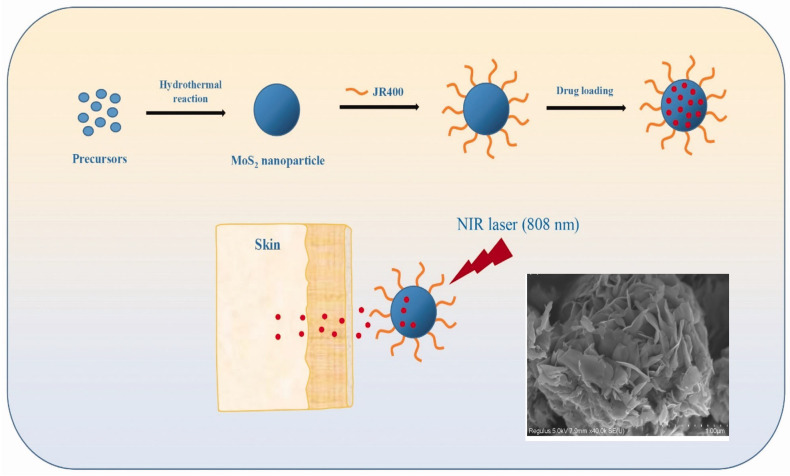
Schematic illustration of the synthesis of JR400-MoS_2_ NPs, drug (ATE) loading, and in vitro skin penetration experiments. Inset: SEM image of one NP [145]. Reproduced with the permission of Taylor and Francis (Copyright 2020).

**Figure 14 micromachines-15-00349-f014:**
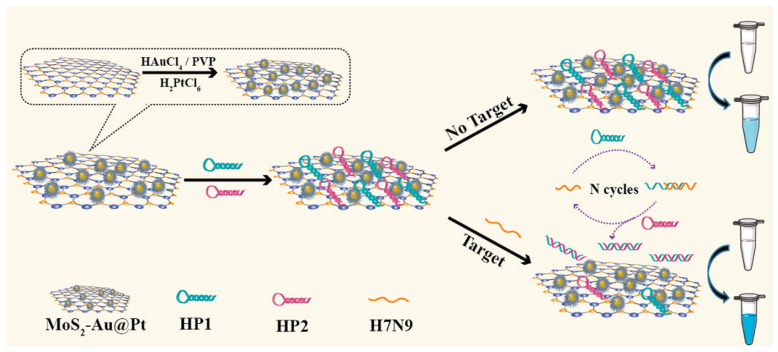
Schematic illustration of a colorimetric sensor based on MoS_2_Au@Pt nanocomposites [146]. Reproduced with the permission of America Chemical Society (Copyright 2022).

**Figure 15 micromachines-15-00349-f015:**
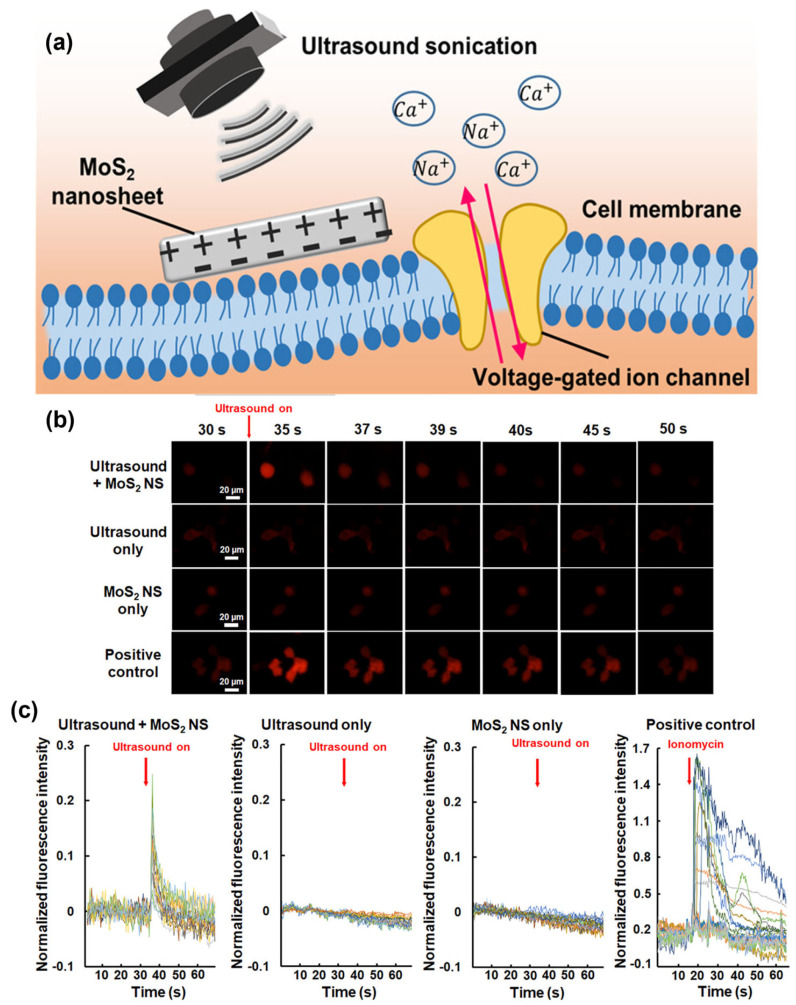
(**a**) Schematic illustration of the underlying mechanism. (**b**) Time-lapse Ca^2+^ images of the neuron cells upon different stimulation conditions. (**c**) Time course of Δ*F*/*F*_0_. Data are obtained from three to six independent experiments [147]. Reproduced with the permission of America Chemical Society (Copyright 2023).

**Figure 16 micromachines-15-00349-f016:**
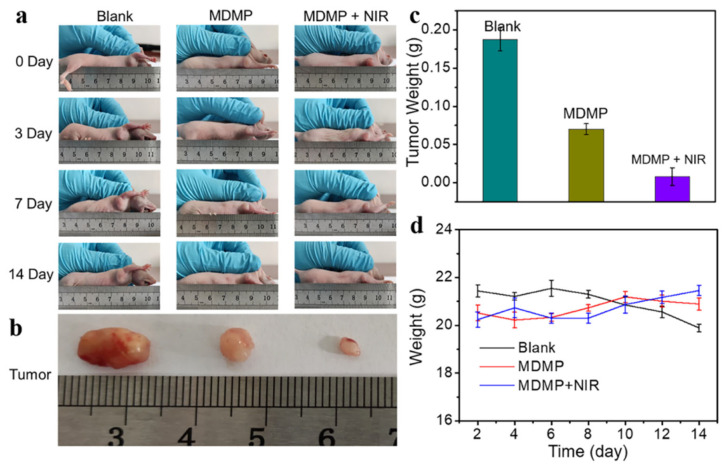
(**a**) Tumor changes during 14 days of PBS, MDMP, and MDMP + NIR treatment. (**b**) Tumor size map after the treatment of PBS, MDMP, and MDMP + NIR. (**c**) Tumor weight after the treatment of PBS, MDMP, and MDMP + NIR. (**d**) Body weight changes of mice during 14 days of treatment by PBS, MDMP, and MDMP + NIR [148]. Reproduced with the permission of America Chemical Society (Copyright 2022).

**Figure 17 micromachines-15-00349-f017:**
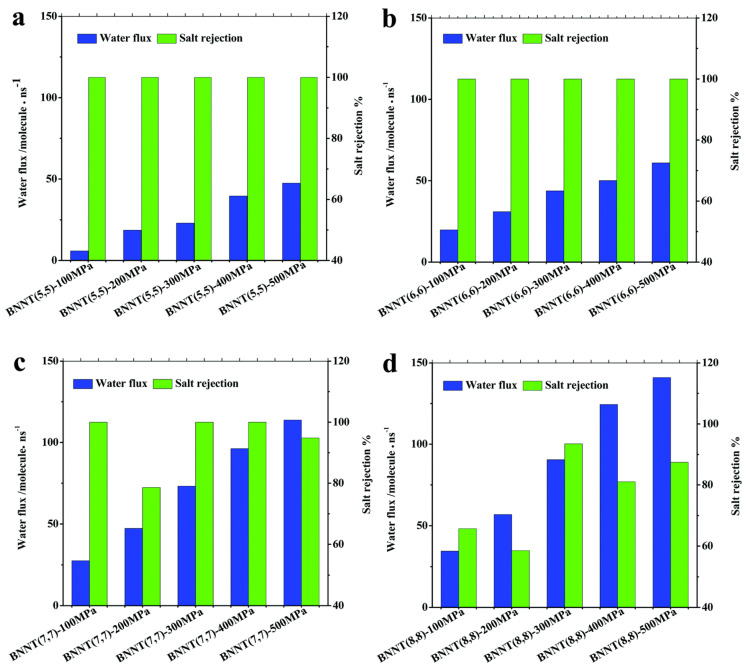
The salt rejection (green bars) and water permeation (blue bars) in BNNT at various chiralities [(5,5) in (**a**), (6,6) in (**b**), (7, 7) in (**c**), and (8,8) in (**d**)] and applied pressures [159]. Reproduced with the permission of the Royal Society of Chemistry (Copyright 2017).

**Figure 18 micromachines-15-00349-f018:**
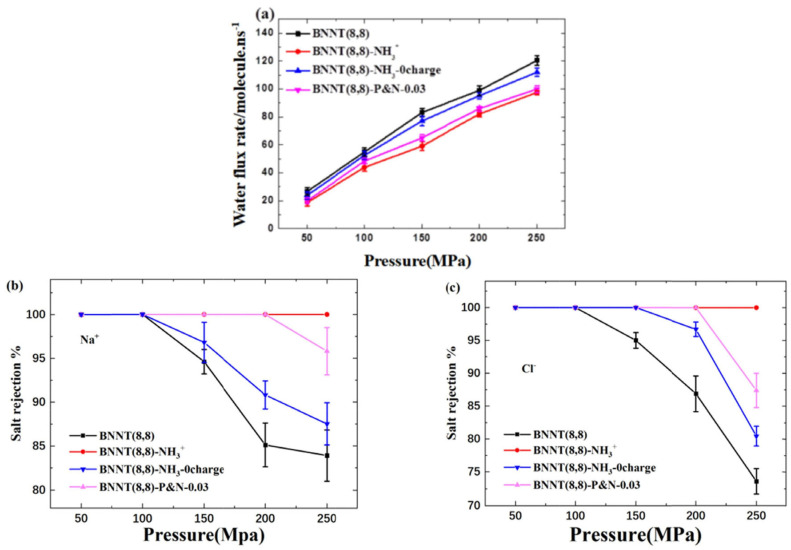
(**a**) Water flux rate and salt rejection of (**b**) Na^+^ and (**c**) Cl^−^ of each system under different pressures [162]. Reproduced with the permission of Elsevier (Copyright 2023).

**Figure 19 micromachines-15-00349-f019:**
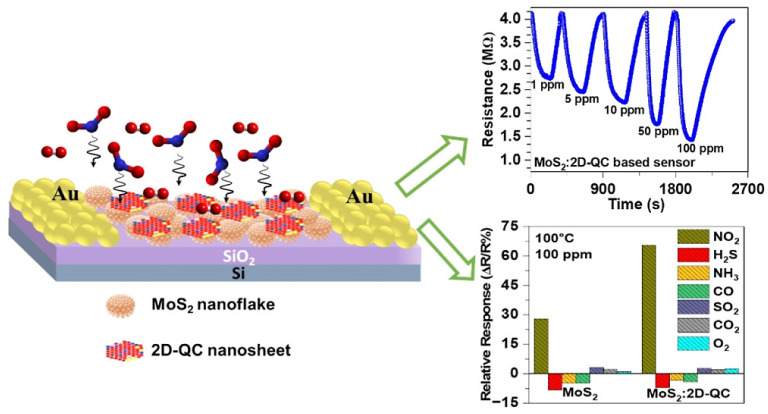
Schematic of the sensor (**left**), transient resistance curves of NO_2_ gas sensing on MoS_2_:2D−QC nanocomposite sensors at 100 °C (**upper right**), and the selectivity responses of MoS_2_ and MoS_2_:2D−QC nanocomposite sensors (**bottom right**) [182]. Reproduced with the permission of America Chemical Society (Copyright 2023).

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
