# Peer review of "Progress in Electronic, Energy, Biomedical and Environmental Applications of Boron Nitride and MoS2 Nanostructures"

_micromachines, 2024, doi:10.3390/mi15030349_

Round 1

Reviewer 1 Report

Comments and Suggestions for Authors

The manuscript titled “Progress in Electronic, Energy, Biomedical and Environmental Applications of Boron Nitride and MoS2 Nanostructures” is a comprehensive review of the latest developments in the application of boron nitride (BN) and molybdenum disulfide (MoS2) nanostructures. It explores their use in various fields including electronics, energy, biomedicine, and environmental science. The article underscores the potential of these nanostructures in advancing technology and improving lives. It provides a valuable resource for researchers and professionals in these fields.

The manuscript is well-written and well-organized, so it is already in good form to be published. I only have several minor suggestions listed below:

1.     In figure 1, it seems the figure shows a topic of pollution. Other aspects of this review, e.g, electronic, biomedical, etc, are not highlighted.

2.     In Fig. 2d, it seems the back gate is suddenly inserted into the hBN. Is this the real case?

3.     Also in Fig. 2, kindly label S, D, G in all relevant figures.

4.     In Fig. 3 a, etc, please label the materials and the structures where it’s not obvious.

5.     For the introduction to electronic applications, it is a pity no figures show their IV, BV, CV, or switching performance.

6.     I suggest introducing more mechanisms and physics into the review paper to make it deeper.

7.     Some figures’ resolution is quite low.

8.     Please consider commenting on the recent progress in MoS2 FET, such as 10.1038/s41586-022-04588-2 and 10.1038/s41928-022-00753-7.

Overall, it is an interesting review paper. Although I listed several items, they are all minor suggestions and optional. The manuscript can be published as it is or after a minor revision.

Author Response

We appreciate all reviewers for their constructive comments and suggestions. Our response is itemized as follows.

Reviewer 1:

The manuscript titled “Progress in Electronic, Energy, Biomedical and Environmental Applications of Boron Nitride and MoS2 Nanostructures” is a comprehensive review of the latest developments in the application of boron nitride (BN) and molybdenum disulfide (MoS2) nanostructures. It explores their use in various fields including electronics, energy, biomedicine, and environmental science. The article underscores the potential of these nanostructures in advancing technology and improving lives. It provides a valuable resource for researchers and professionals in these fields.

The manuscript is well-written and well-organized, so it is already in good form to be published. I only have several minor suggestions below:

Thank you for your supportive decision.

1.In figure 1, it seems the figure shows a topic of pollution. Other aspects of this review, e.g, electronic, biomedical, etc, are not highlighted.

We have edited Figure 1 with images of electronics, lithium-ion batteries, and DNAs to match the text labeling.

2.In Fig. 2d, it seems the back gate is suddenly inserted into the hBN. Is this the real case?

The figure is adopted from reference 29 without modification. The h-BN is laid on top of the Au electrode.

3.Also in Fig. 2, kindly label S, D, G in all relevant figures.

The figure (now Figure 3 as we added a new Figure 2) is adopted from reference 29, and we prefer not to make any modifications.

4.In Fig. 3 a, etc, please label the materials and the structures where it’s not obvious.

We prefer to use the figure caption to describe the figure, just so that the graphic will not be overpowered with text.

5.For the introduction to electronic applications, it is a pity no figures show their IV, BV, CV, or switching performance.

We agreed that such figures are helpful (we have some in figures 3, 5, 6). Given our broad topic coverage, we prefer to attract readers with a description of the research and conclusion and refer them to read the details in the original references. 

6.I suggest introducing more mechanisms and physics into the review paper to make it deeper.

We agreed, and we wish to do so. However, this will change the review article's flavor to introduce readers to recent research progress in broad topic coverage. The review allows readers to capture the big picture of recent progress and refers them to read the details in the original references. 

7.Some figures’ resolution is quite low.

We ensured that we used the high-resolution images downloaded from the original reference. Figures 10, 11 are now improved in resolution.

8.Please consider commenting on the recent progress in MoS2 FET, such as 10.1038/s41586-022-04588-2 and10.1038/s41928-022-00753-7.

Thank you for the great suggestion. We have added a paragraph (after Figure 7) to introduce these two new references (77, 78).

Overall, it is an interesting review paper. Although I listed several items, they are all minor suggestions and optional. The manuscript can be published as it is or after a minor revision.

Thank you for your supportive decision.

Additional revision:

  • The reproduction of the original Figure 6 (now Figure 7 after adding a new Figure 2) requires payment to Nature Journal. We replaced it with an image from MIT’s news release (Public domain) and modified the text description accordingly.
  • The English of the manuscript has been further edited.

Reviewer 2 Report

Comments and Suggestions for Authors

The manuscript (micromachines-2868510) described the recent development of BN and MoS2 for a wide range of applications. The topic is interesting and the research on this realm is a recent hot topic. Before the consideration of the acceptance, the following comments should be addressed.

Comment 1: Why put BN and MoS2 together in this review, since there are many differences between these two materials although they are both 2D material. Please state more about the logic for chosing BN and MoS2 in the introduction part.

Comment 2: The structure and physicochemical properties for the 2D material (BN, MoS2)  should be in-detail analyzed, which will offer the foundamental for the application.

Comment 3: Please pay attention to the format of the Figures, for example, there is lines under Figire 1c and f. Please carefully check the whole Figures and make modifications.

Comment 4: The permission of the Figures from the publication press should be offered maybe after the acceptance of this manuscript.

Comment 5: A comparison of BN with other commonly adapted material is suggested to be included in the applications such as solar cells and others.

Comment 6: There are also format mistakes all through the whole manuscript such as oC in Line 207, h-BN rather than hBN in Line 211. There is background in Line 364-367. A g-1 in Line 371. Please carefully check. 

Comment 7: For the application of MoS2, the hydrogen production from electronic hydrgen evolution (HER) reaction or orther reaction should not be excluded. The following manuscript is suggested to be cited: Chem. Commun., 2019, 55, 628--631, Journal of Power Sources 431 (2019) 135–143, Applied Catalysis B: Environmental 314 (2022) 121495, Materials Horizons, 2024, DOI: 10.1039/D3MH01909H. 

Comment 8: Please improve the quality of the Fig. 10. The author could download the original image from the journal.

Comment 9: For the enviromental application, it will be more intriguing to total abetement  of the air pollutant compared to physically trapping. Therefore BN will be a good catalyst support for noble metal catalyst or metal oxide catalyst. Please make short comments and analysis on this part. The following manuscript is suggested for the citation: Molecular Catalysis 553 (2024) 113768, Applied Catalysis B: Environmental 272 (2020) 118858, Nature Commun.2017, 8, 15291

Comment 10: It is strongly suggest the author to analyze more about the structure-function relationship in each part to clarify the superiority of BN and MoS2 with other materials. 

Comment 11: The further development direction should be described specificially, e.g. by what method the performance could be improved.

Comments on the Quality of English Language

The English is fine.

Author Response

We appreciate all reviewers for their constructive comments and suggestions. Our response is itemized as follows.

Reviewer 2:

The manuscript (micromachines-2868510) described the recent development of BN and MoS2 for a wide range of applications. The topic is interesting and the research on this realm is a recent hot topic. Before the consideration of the acceptance, the following comments should be addressed.

Comment 1: Why put BN and MoS2 together in this review, since there are many differences between these two materials although they are both 2D material. Please state more about the logic for chosing BN and MoS2 in the introduction part.

As written in the introduction, our manuscript focuses on nanostructures with sizable bandgaps. This allows us to focus on BN and MoS2, not graphene, MXene, and carbon nanotubes. This scope of work is already quite extensive. Therefore, we only focus on one type of TMDCs, i.e. MoS2. We added a statement in the abstract to highlight the focus on materials with sizable bandgaps.

Comment 2: The structure and physicochemical properties for the 2D material (BN, MoS2) should be in-detail analyzed, which will offer the foundamental for the application.

We agreed that this can be convenient for readers by including the structure and physicochemical properties. Since the title and scope of this manuscript are about application, we prefer to refer readers to other review articles for the properties. A statement (with references) is now added at the end of the introduction section, along with a new Figure 2 that illustrates the structures of BN and MoS2 nanostructures.

Comment 3: Please pay attention to the format of the Figures, for example, there is lines under Figire 1c and f. Please carefully check the whole Figures and make modifications.

Thank you for your careful reading. Some of those may also come from pdf conversion, as we didn’t see them in our Word document.

Comment 4: The permission of the Figures from the publication press should be offered maybe after the acceptance of this manuscript.

Thank you for your careful attention. This is now being added.

Comment 5: A comparison of BN with other commonly adapted material is suggested to be included in the applications such as solar cells and others.

Thank you for the interesting idea. It can be challenging to make such comparisons accurately as those were performed using different device structures and methods.

Comment 6: There are also format mistakes all through the whole manuscript such as oC in Line 207, h-BN rather than hBN in Line 211. There is background in Line 364-367. A g-1 in Line 371.

Please carefully check.

Thank you for your careful attention. We have corrected those you have suggested and several others.

Comment 7: For the application of MoS2, the hydrogen production from electronic hydrgen evolution (HER) reaction or orther reaction should not be excluded. The following manuscript is suggested to

be cited: Chem. Commun., 2019, 55, 628--631, Journal of Power Sources 431 (2019) 135–143, Applied Catalysis B: Environmental 314 (2022) 121495, Materials Horizons, 2024, DOI:10.1039/D3MH01909H.

Thank you for the great suggestion. We have added a new paragraph with information regarding hydrogen evolution reactions (HER) and water splitting in section 2.2.3. We have included some of the relevant suggested references, among others.

Comment 8: Please improve the quality of the Fig. 10. The author could download the original image from the journal.

We ensured that we used the high-resolution images downloaded from the original reference. Figures 10, 11 are now improved in resolution.

Comment 9: For the enviromental application, it will be more intriguing to total abetement of the air pollutant compared to physically trapping. Therefore BN will be a good catalyst support for noble metal catalyst or metal oxide catalyst. Please make short comments and analysis on this part. The following manuscript is suggested for the citation: Molecular Catalysis 553 (2024) 113768, Applied Catalysis B: Environmental 272 (2020) 118858, , Nature Commun., 2017, 8, 15291

There are errors in the suggested citation. We could not find the first two suggested references. We have discussed the 3rd suggested reference at the end of section 4.1.1.

Comment 10: It is strongly suggest the author to analyze more about the structure-function relationship in each part to clarify the superiority of BN and MoS2 with other materials.

We agreed that this can be convenient for readers by including the structure and physicochemical properties. Since the title and scope of this manuscript are about application, we prefer to refer readers to other review articles for the properties. A statement (with references) is now added at the end of the introduction section.

Comment 11: The further development direction should be described specificially, e.g. by what method the performance could be improved.

Given the broad scope of work reviewed in this manuscript, it is impossible to expressly point out what to improve regarding methods for each application. We think it best to encourage researchers to explore their creativity in the two major trends pointed out in section 5.

Additional revision:

  • The reproduction of the original Figure 6 (now Figure 7 after adding a new Figure 2) requires payment to Nature Journal. We replaced it with an image from MIT’s news release (Public domain) and modified the text description accordingly.
  • The English of the manuscript has been further edited.

Reviewer 3 Report

Comments and Suggestions for Authors

Having thoroughly read the article titled "Progress in Electronic, Energy, Biomedical, and Environmental Applications of Boron Nitride and MoS2 Nanostructures," I found it to be highly insightful and relevant. However, I recommend that the publication of this work be considered after the authors address the following significant details:

1.      Authors are encouraged to include a comparative table illustrating recent advancements in the various applications of boron nitride and MoS2 nanostructures.

2.      In the introduction section, authors are encouraged to discuss other nanomaterials and cite the following references: a) Scalable functionalized liquid crystal elastomer fiber soft actuators with multi-stimulus responses and photoelectric conversion, Mater. Horiz., 2023, 10, 2587. b) Facile optimization of hierarchical topography and chemistry on magnetically active graphene oxide nanosheets, Chemical Science 11 (25), 6556-6566.

3.      Boron nitride-based nanomaterials have also been utilized in water splitting applications. Please provide a discussion on this topic.

4.      Authors are encouraged to discuss the various morphological shapes of BN nanomaterials, including nanospheres, nanotubes, nanofibers, nanoribbons, thin films, and nanosheets. An additional figure illustrating these morphologies would be beneficial.

5.      Why is MoS2 highly active? The strain energy of molybdenum disulfide significantly exceeds the energies of carbon and many other similar compounds. Please include this discussion.

6.      Please discuss the photocatalytic mechanism of MoS2 in Section 4.2.1.

Comments on the Quality of English Language

Minor editing of English language required

Author Response

We appreciate all reviewers for their constructive comments and suggestions. Our response is itemized as follows.

Reviewer 3:

Having thoroughly read the article titled "Progress in Electronic, Energy, Biomedical, and Environmental Applications of Boron Nitride and MoS2 Nanostructures," I found it to be highly insightful and relevant. However, I recommend that the publication of this work be considered after the authors address the following significant details:

Thank you for your supportive decision.

  1. Authors are encouraged to include a comparative table illustrating recent advancements in the various applications of boron nitride and MoS2 nanostructures.

We agreed and wished that we could create one. Unfortunately, we are reviewing the recent advancements reported in two different materials, and it is impossible to match and compare identical applications between BN and MoS2. Thank you for your understanding.

2.In the introduction section, authors are encouraged to discuss other nanomaterials and cite the following references: a) Scalable functionalized liquid crystal elastomer fiber soft actuators with multi-stimulus responses and photoelectric conversion, Mater. Horiz.,2023, 10, 2587. b) Facile optimization of hierarchical topography and chemistry on magnetically active graphene oxide nanosheets, Chemical Science 11 (25), 6556-6566.

Thank you for suggesting the wonderful references. These two publications are about MXene and graphene oxide. Since our focus is BN and MoS2, it is not appropriate to discuss these nanomaterials in detail.

3.Boron nitride-based nanomaterials have also been utilized in water splitting applications. Please provide a discussion on this topic.

Thank you for your suggestion. We have added a new (second) paragraph in section 2.1.1. on HER and water splitting.

4.Authors are encouraged to discuss the various morphological shapes of BN nanomaterials, including nanospheres, nanotubes, nanofibers, nanoribbons, thin films, and nanosheets. An additional figure illustrating these morphologies would be beneficial.

We agreed that this can be convenient for readers by including the structure and physicochemical properties. Since the title and scope of this manuscript are about application, we prefer to refer readers to other review articles for the properties. A statement (with references) is now added at the end of the introduction section, along with a new Figure 2 that illustrates the appearance of these nanostructures.

5.Why is MoS2 highly active? The strain energy of molybdenum disulfide significantly exceeds the energies of carbon and many other similar compounds. Please include this discussion.

We did not mention that MoS2 is highly active. We are summarizing the published work, which cites that the edges of MoS2 are catalytically/electrochemically active.

6.Please discuss the photocatalytic mechanism of MoS2 in Section 4.2.1.

Thank you for raising this great question. As we have discussed for reference 172, MoS2 work as “a direct photosensitizer which significantly enhanced the photocatalytic activity under visible light by facilitating efficient electron transfer from nano-MoS2 to TiO2­, …”. In the discussion of reference 173, “Photodegradation tests demonstrated that rGO/MoS2 outperformed pure MoS­2, rGO, and TiO2 in degrading methylene blue in visible light.” Therefore, the enhanced photocatalytic effect of these hybridized materials is not due to MoS­2 alone. It is the combined properties of two materials that improve the performance. A statement is now added to reiterate this fact.

Additional revision:

  • The reproduction of the original Figure 6 (now Figure 7 after adding a new Figure 2) requires payment to Nature Journal. We replaced it with an image from MIT’s news release (Public domain) and modified the text description accordingly.
  • The English of the manuscript has been further edited.

Round 2

Reviewer 2 Report

Comments and Suggestions for Authors

The reviewer has well addressed the comments, the manuscript is suggested to be accepted without any further modifications.

Reviewer 3 Report

Comments and Suggestions for Authors

All the comments have been adequately addressed. This paper is suitable for publication.